



# Multiscale modeling of heat and mass transfer in dry snow: influence of the condensation coefficient and comparison with experiments

Lisa Bouvet[1,2], Neige Calonne[1], Frédéric Flin[1], and Christian Geindreau[2]

[1]Univ. Grenoble Alpes, Université de Toulouse, Météo-France, CNRS, CNRM, Centre d'Études de la Neige, Grenoble, France
[2]Université Grenoble Alpes, CNRS, Grenoble INP, 3SR, Grenoble, France

**Correspondence:** Lisa Bouvet (lisa.bouvet@univ-grenoble-alpes.fr)

**Abstract.** Temperature gradient metamorphism in dry snow is driven by heat and water vapor transfer through snow, which includes conduction/diffusion processes in both air and ice phases as well as sublimation and deposition at the ice-air interface. The latter processes are driven by the condensation coefficient $\alpha$, a poorly constrained parameter in literature. In the present paper, we use an upscaling method to derive heat and mass transfer models at the snow layer scale according to $\alpha$ in the range $10^{-10}$ to 1. A transition $\alpha$-value arises, of the order of $10^{-4}$ for typical snow microstructures (characteristic length $\sim 0.5$ mm), such as the vapor transport is limited by sublimation-deposition below that value and by diffusion above. Accordingly, different macroscopic models with specific domains of validity with respect to $\alpha$-values are derived. A comprehensive evaluation of the models is presented by comparing with three experimental datasets as well as with pore-scale simulations using a simplified microstructure. The models reproduce the two main features of the experiments: the non-linear temperature profiles, with enhanced values in the center of the snow layer, and the mass transfer, with an abrupt basal mass loss. However, both features are overall underestimated by the models when compared to the experimental data. We investigate possible causes of these discrepancies and suggest potential improvements for the modeling of heat and mass transport in dry snow.

## 1 Introduction

Natural snowpacks are frequently subjected to temperature gradients induced by the meteorological conditions. In case of temperature gradient in dry snow, heat and water vapor are transported through the snowpack by heat conduction through ice and air and by vapor diffusion in air. These phenomena are coupled by the sublimation-deposition processes at the ice-air interfaces. In practice, such transfer processes can be enhanced by natural air convection induced by the temperature gradient (e.g., Jafari et al., 2022) or by forced convection generated by the wind at the snowpack surface (e.g., Albert and McGilvary, 1992; Calonne et al., 2015). For a sake of simplicity, both types of convection are neglected in the following. All those processes lead to changes in the snow microstructure called temperature gradient metamorphism (TGM), which transforms snow into faceted crystals (FC) in the case of moderate gradients and into depth hoar (DH) for stronger gradients (see Fierz et al., 2009). Those transformations of the microstructure can sometimes come along with a redistribution of mass in the snow layer, a density





drop or even the formation of an air gap at the base of the snowpack, as observed in the Arctic (e.g., Domine et al., 2019) or in some cold room experiments (e.g., Kamata and Sato, 2007; Wiese, 2017; Bouvet et al., 2023). As a result of changes

in microstructure and density, TGM also induces significant changes in the snow physical and mechanical properties, such as thermal conductivity, vapor diffusivity, or elastic properties (e.g., Srivastava et al., 2010; Calonne et al., 2014a; Wautier et al., 2015), affecting the snowpack behavior at larger scale. Hence, an accurate representation of the heat and mass transport processes during TGM is key to accurately model the snow cover, as required for many applications as avalanche forecasting or climate studies (Jordan, 1991; Lehning et al., 2002; Vionnet et al., 2012).

In micro-scale modeling, heat and mass transfer processes are coupled through boundary conditions accounting for sublimation and deposition processes at the air-ice interface and involving the interface growth velocity. The latter is classically given by the Hertz-Knudsen equation, which is linearly dependent on the condensation coefficient $\alpha$. This parameter, also called condensation, attachment, sticking, deposition, or kinetic coefficient (e.g., Flin et al., 2003; Libbrecht, 2005; Brzoska et al., 2008; Kaempfer and Plapp, 2009; Furukawa, 2015; Krol and Löwe, 2016; Fourteau et al., 2021a; Granger et al., 2021), theoretically

ranges from 0 to 1 for an infinite flat surface, since it characterizes the probability that a water vapor molecule striking the ice surface is incorporated into it (see e.g., Libbrecht, 2005; Furukawa, 2015). The analogous coefficient for sublimation can also be defined, although it is classically assumed equal to the condensation coefficient. However, this coefficient is still poorly understood and quantified, notably because of its complex dependencies on temperature, temperature gradient, supersaturation, and crystalline orientation (see, e.g., Libbrecht, 2021). Values of the condensation coefficient can be found in the literature,

from single crystal growth experiments, usually ranging from $10^{-4}$ to $10^{-1}$ (see, e.g., Libbrecht and Rickerby, 2013), or indirectly retrieved from snow modeling at the pore scale, ranging from $10^{-4}$ to $10^{-3}$ (see, e.g., Flin, 2004; Bouvet et al., 2022). Currently, modeling heat and mass transport at the pore scale can only be performed on small snow volumes (e.g., Kaempfer and Plapp, 2009; Vetter et al., 2010; Bouvet et al., 2022). To predict the behavior of the entire snowpack, macro-scale models, i.e. at the scale of the snow layer, are used.

In the last decades, several models have been proposed to describe heat and mass transfer at the snow layer scale. The first models assumed saturated vapor conditions in the snow (e.g., de Quervain, 1963; Anderson, 1976; Powers et al., 1985). Later, using a phenomenological approach, Albert and McGilvary (1992) proposed to describe the heat and water vapor transfer through a snowpack subjected to an air flow, without restricting the water vapor to its saturation value. The model uses two coupled advection-diffusion equations including a source term arising from phase change at the pore scale. A similar heat and

mass transfer model was analytically obtained by Calonne et al. (2014b, 2015) using an upscaling method. In that case, the macroscopic equivalent modeling was derived from its description at the pore scale using the homogenization of multiple scale expansions. This theoretical method also provides the exact expression of the effective parameters arising at the macro-scale and the domains of validity of the macroscopic modeling. Two main effective parameters emerge from the model: (i) the effective thermal conductivity $k^{\text{eff}}$, which depends on the ice and air conductivity and on the snow microstructure, and (ii) the

effective diffusion $D^{\text{eff}}$, which depends on the vapor molecular diffusion coefficient and the snow microstructure. The source term is related to the Hertz-Knudsen equation, which involves the condensation coefficient $\alpha$. Calonne et al. (2014b) have shown that this model is valid for interface growth velocities below $3 \times 10^{-11}$ m s$^{-1}$, which typically corresponds to slow



kinetics.

Other approaches largely rely on the assumption of saturated vapor conditions, which seems valid for faster kinetics and rather
high values of $\alpha$ (e.g., Sturm and Benson, 1997; Kamata and Sato, 2007; Hansen and Foslien, 2015). Hansen and Foslien (2015)
developed a heat and mass transfer model using a mixture theory. Assuming that the water vapor is saturated (based on the value
of $\alpha = 0.0144$ from Delaney et al. (1964)), the authors derived a unique thermal equation which yields an apparent thermal
conductivity that depends on the air and ice conductivities, the water vapor diffusivity, the latent heat of sublimation-deposition,
and the temperature derivative of the Clapeyron equation. A similar formulation of the apparent thermal conductivity was also
proposed by Yosida et al. (1955). Recently, Fourteau et al. (2021b) investigated the influence of $\alpha$ on the apparent diffusion
coefficient in snow. By performing numerical simulation on 3D images, they showed that this apparent diffusion coefficient
is equal to $D^{\text{eff}}$ for $\alpha$-values smaller than $\approx 10^{-4}$ and then increases with increasing $\alpha$ until it reaches a plateau for $\alpha$ larger
than $10^{-2}$; the value at the plateau being smaller than the molecular diffusion of water vapor in the air. In a companion paper,
Fourteau et al. (2021a) computed from 3D images the apparent thermal conductivity of snow assuming that the water vapor
on the ice-air interface is equal to the water vapor at saturation given by the Clapeyron equation. In this case, they showed that
the apparent thermal conductivity is enhanced by the sublimation-deposition process arising at the pore scale. Their results are
consistent with the model of Moyne et al. (1988) for wet porous media based on the same hypothesis at the micro-scale and
derived using the volume averaging method.

Further uses of the above mentioned models, as their implementation in full snow cover models, are limited by some chal-
lenges. One is the difficulty of choosing between models as they differ in many ways: they were derived using different
methods, involve different balance equations and effective parameters, and are valid for different, often unclear, domains of
validity in terms of $\alpha$-values. This should be clarified, especially by estimating the $\alpha$-values from which the assumption of
saturated water vapor is theoretically valid. A second challenge is that none of these models were thoroughly evaluated to as-
sess their performances. This might be partly due to the limited number of suited datasets to compare with. The datasets from
the cold-laboratory experiments of Kamata and Sato (2007) and, recently, of Bouvet et al. (2023) seem however relevant for
such comparisons, as they provide time-series of the vertical profiles of snow density and temperature, as well as the forcing
conditions to be reproduced in the simulations.

This paper aims i/ to define the heat and mass transport modeling in dry snow for the full $\alpha$-values range and ii/ to evaluate the
model's ability to reproduce natural snow evolution during TGM. To this end, in a first part, the homogenization of multiple
scale expansions is applied to derive the macroscopic equivalent modeling of heat and vapor transfer for $\alpha$-values ranging
from $10^{-10}$ to 1, following Calonne et al. (2014b). The physics considered at the pore scale includes heat conduction, vapor
diffusion, and phase change; neglecting any transport linked to curvature effect and convection. The macroscopic models and
the involved macroscopic properties are compared to the ones from the literature and illustrated for simplified snow microstruc-
tures. In a second part, the derived macroscopic models are evaluated using three cold-laboratory experiments of TGM from
Kamata and Sato (2007) and Bouvet et al. (2023). The experiments are reproduced with the macroscopic models and results
between observations and simulations are analyzed.




## 2 Derivation of the macroscopic modeling

### 2.1 Upscaling method

We apply the homogenization technique of multiple scale expansion (Bensoussan et al., 1978; Sanchez-Palencia, 1980) to the

physics of heat and vapor transport in dry snow. The homogenization method allows to model the local physical processes in heterogeneous media by an equivalent continuous macroscopic description if the condition of separation of scales is satisfied (Bensoussan et al., 1978; Sanchez-Palencia, 1980; Auriault, 1991; Auriault et al., 2009). This coefficient of separation of scales can be expressed as $\varepsilon = l/L \ll 1$, where $l$ and $L$ are the characteristic lengths of the heterogeneities at the pore scale and of the macroscopic sample or excitation, respectively. This condition implies the existence of a Representative Elementary

Volume (REV) of size $l$ for both the material and the excitation. Following the methodology presented by Auriault (1991), the macroscopic equivalent model is obtained from the description of the physics at the pore scale by: (i) assuming the medium to be periodic, without loss of generality as the condition $\varepsilon = l/L \ll 1$ is fulfilled; (ii) writing the description of the physics at the pore scale in a dimensionless form; (iii) evaluating the obtained dimensionless numbers with respect to the coefficient of separation of scale $\varepsilon$; (iv) looking for the unknown fields in the form of asymptotic expansions in powers of $\varepsilon$; and (v) solving

the successive boundary-value problems that are obtained after introducing these expansions in the pore scale dimensionless description. The macroscopic equivalent model is obtained from compatibility conditions that are the necessary conditions for the existence of solutions to the boundary-value problems.

### 2.2 Physical processes at the pore scale

As in Calonne et al. (2014b), we assume that a snow layer of characteristic length $L$ can be represented by a collection of spatially periodic REVs of characteristic length $l$ such that the coefficient of separation of scale $\varepsilon = l/L \ll 1$. In what follows,

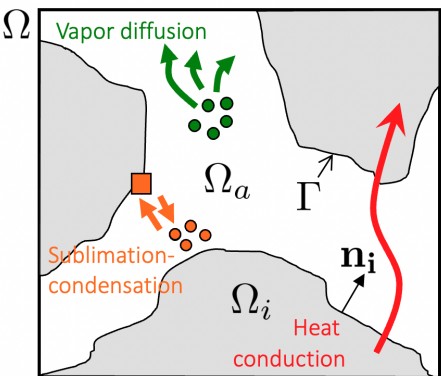

**Figure 1.** Physical phenomena under consideration at the Representative Elementary Volume (REV) scale.


$\Omega$ is the REV domain, $\Omega_i$ is the ice domain, and $\Omega_a$ is the air domain (Fig. 1). The ice grains interface is noted $\Gamma$ and $\mathbf{n_i}$ is the unit outward vector of $\Omega_i$. The subscripts $(_i)$ or $(_a)$ are related to quantities defined in $\Omega_i$ and $\Omega_a$, respectively. As illustrated in





Fig. 1, the processes of heat and mass transport in dry snow considered are (i) the heat conduction through ice and air, (ii) the water vapor diffusion in air, and (iii) the sublimation of ice and deposition of vapor at the ice grain interface, characterized by

an interface growth velocity (Libbrecht, 2005; Kaempfer and Plapp, 2009; Barrett et al., 2012) following the Hertz-Knudsen equation. Air convection and snow densification are not taken into account here. Assuming that the properties of air and ice are isotropic, these physical processes at the pore scale are described by the following set of equations:

$$\rho_i C_i \frac{\partial T_i}{\partial t} - \operatorname{div}(k_i \mathbf{grad} T_i) = 0 \quad \text{in } \Omega_i \tag{1}$$

$$\rho_a C_a \frac{\partial T_a}{\partial t} - \operatorname{div}(k_a \mathbf{grad} T_a) = 0 \quad \text{in } \Omega_a \tag{2}$$

$$\frac{\partial \rho_v}{\partial t} - \operatorname{div}(D_v \mathbf{grad} \rho_v) = 0 \quad \text{in } \Omega_a \tag{3}$$

$$T_i = T_a \quad \text{on } \Gamma \tag{4}$$

$$k_i \mathbf{grad} T_i \cdot \mathbf{n_i} - k_a \mathbf{grad} T_a \cdot \mathbf{n_i} = L_{sg} \mathbf{w} \cdot \mathbf{n_i} \quad \text{on } \Gamma \tag{5}$$

$$D_v \mathbf{grad} \rho_v \cdot \mathbf{n_i} = (\rho_i - \rho_v) \mathbf{w} \cdot \mathbf{n_i} \simeq \rho_i \mathbf{w} \cdot \mathbf{n_i} \quad \text{on } \Gamma \tag{6}$$

where $t$ is the time (s), $T$ is the temperature (K), $k$ is the thermal conductivity (W m$^{-1}$ K$^{-1}$), $\rho$ is the density (kg m$^{-3}$), $C$ is the

specific heat capacity (J kg$^{-1}$ K$^{-1}$), $L_{sg}$ is the latent heat of sublimation-deposition (J m$^{-3}$), $\mathbf{w}$ is the interface growth velocity (m s$^{-1}$), $\rho_v$ is the partial density of water vapor in air (kg m$^{-3}$), $D_v$ is the water vapor diffusion coefficient in air (m$^2$ s$^{-1}$) and, div and $\mathbf{grad}$ are the divergence and gradient operators with respect to the physical space variable $\mathbf{X}$ respectively. At the interface, the heat and mass transfer are coupled through the normal interface growth velocity $w_n = \mathbf{w} \cdot \mathbf{n_i}$, which is given by the Hertz-Knudsen equation,

$$w_n = \mathbf{w} \cdot \mathbf{n_i} = \frac{1}{\beta} \left[ \frac{\rho_v - \rho_{vs}(T_a)}{\rho_{vs}(T_a)} - d_0 K \right] \quad \text{on } \Gamma \tag{7}$$

such as $w_n$ is positive when the ice grain grows and negative when it sublimates. $\beta$ is the interface kinetic coefficient (s m$^{-1}$), $\rho_{vs}$ is the saturation water vapor density in air (kg m$^{-3}$), $d_0$ is the capillary length (m), and $K$ is the interface mean curvature (m$^{-1}$). The interface kinetic coefficient $\beta$ is linked to the condensation coefficient $\alpha$ by

$$\frac{1}{\beta} = \alpha \frac{\rho_{vs}(T_a)}{\rho_i} \sqrt{\frac{k_B T_a}{2\pi m}} \tag{8}$$

where $m$ is the mass of a water molecule (kg) and $k_B$ is the Boltzmann's constant equal to $1.38 \times 10^{-23}$ J K$^{-1}$. As already mentioned, the condensation coefficient $\alpha$ characterizes the probability for a water molecule hitting the surface of the solid to be incorporated to the crystal, or inversely, and ranges from 0 to 1. Although this coefficient depends on several parameters as





temperature, supersaturation, and crystalline orientation, we assume that this parameter is constant over the REV at first order. The saturation vapor density $\rho_{vs}$ at a given air temperature $T_a$ is given by the Clausius Clapeyron's law

$$\rho_{vs}(T_a) = \rho_{vs}^{\text{ref}}(T^{\text{ref}}) \exp\left[\frac{L_{sg}m}{\rho_i k_B}\left(\frac{1}{T^{\text{ref}}} - \frac{1}{T_a}\right)\right] \tag{9}$$

For simplicity, we assume that none of the material properties ($\rho$, $C$, $k_B$, $D_v$, $\beta$, $m$) depend on the temperature. Also, the effect of curvature on the ice interface growth is considered insignificant compared to the effect of temperature and is neglected. Consequently, using Eq. (8), the Hertz-Knudsen equation can be rewritten

$$w_n = \mathbf{w} \cdot \mathbf{n_i} = \frac{1}{\beta \rho_{vs}(T_a)}\left[\rho_v - \rho_{vs}(T_a)\right] = \frac{\alpha}{\rho_i}w_k(T_a)\left[\rho_v - \rho_{vs}(T_a)\right] \quad \text{on } \Gamma \tag{10}$$

where $w_k = \sqrt{k_B T_a/2\pi m}$ is defined as a kinetic velocity which depends on the temperature at the ice-air interface. Taking into account this result, Eq. (5) and (6) can be rewritten:

$$k_i \mathbf{grad}T_i \cdot \mathbf{n_i} - k_a \mathbf{grad}T_a \cdot \mathbf{n_i} = L_{sg}\frac{\alpha}{\rho_i}w_k(T_a)\left[\rho_v - \rho_{vs}(T_a)\right] = L_{sg}\frac{D_v}{\rho_i}\mathbf{grad}\rho_v \cdot \mathbf{n_i} \quad \text{on } \Gamma \tag{11}$$

$$D_v \mathbf{grad}\rho_v \cdot \mathbf{n_i} = \alpha w_k(T_a)\left[\rho_v - \rho_{vs}(T_a)\right] \quad \text{on } \Gamma \tag{12}$$

## 2.3 Dimensionless pore scale description

The next step is the normalization of the above pore scale description Eq. (1) - (4) and (11) - (12). For that, all the dimensional variables in this description are written such as each variable $\varphi$ reads $\varphi = \varphi_c \varphi^*$, where the subscript 'c' denotes a characteristic quantity (constant) and the superscript '*' denotes a dimensionless variable. Note that the microscopic length $l$ is chosen as characteristic length such as $l_c = l$, i.e. the so-called microscopic point of view is adopted (Auriault, 1991). The formal dimensionless set of equations that describes the physics at the pore scale can thus be written as:

$$\left[F_i^T\right]\rho_i^* C_i^* \frac{\partial T_i^*}{\partial t^*} - \text{div}^*(k_i^* \mathbf{grad}^* T_i^*) = 0 \quad \text{in } \Omega_i \tag{13}$$

$$\left[F_a^T\right]\rho_a^* C_a^* \frac{\partial T_a^*}{\partial t^*} - \text{div}^*(k_a^* \mathbf{grad}^* T_a^*) = 0 \quad \text{in } \Omega_a \tag{14}$$

$$\left[F_a^\rho\right]\frac{\partial \rho_v^*}{\partial t^*} - \text{div}^*(D_v^* \mathbf{grad}^* \rho_v^*) = 0 \quad \text{in } \Omega_a \tag{15}$$

$$T_i^* = T_a^* \quad \text{on } \Gamma \tag{16}$$

$$[K]\,k_i^* \mathbf{grad}^* T_i^* \cdot \mathbf{n_i} - k_a^* \mathbf{grad}^* T_a^* \cdot \mathbf{n_i} = [H]\,L_{sg}^* \frac{D_v^*}{\rho_i^*}\mathbf{grad}^* \rho_v^* \cdot \mathbf{n_i} \quad \text{on } \Gamma \tag{17}$$

$$D_v^* \mathbf{grad}^* \rho_v^* \cdot \mathbf{n_i} = [W_R]\,\alpha^* w_k^* \left[\rho_v^* - [R]\,\rho_{vs}^*(T_a^*)\right] \quad \text{on } \Gamma \tag{18}$$





This dimensionless description introduces seven dimensionless numbers that characterize the relative intensity of the physical processes at the pore scale. These dimensionless numbers are defined as:

$$\left[ \mathrm{F}_i^T \right] = \frac{l^2 \rho_{i_c} C_{i_c}}{t_c k_{i_c}}, \quad \left[ \mathrm{F}_a^T \right] = \frac{l^2 \rho_{a_c} C_{a_c}}{t_c k_{a_c}}, \quad \left[ \mathrm{F}_a^\rho \right] = \frac{l^2}{D_{v_c} t_c}, \quad [\mathrm{K}] = \frac{k_{i_c}}{k_{a_c}}, \quad [\mathrm{W_R}] = \frac{l \alpha_c w_{k_c}}{D_{v_c}}, \quad [\mathrm{R}] = \frac{\rho_{vs_c}(T_{a_c})}{\rho_{v_c}},$$


$$[\mathrm{H}] = \frac{l L_{sg_c} w_{n_c}}{k_{a_c} T_{a_c}} \quad \text{with} \quad w_{n_c} = \frac{\alpha_c w_{k_c}}{\rho_{i_c}}(\rho_{v_c} - \rho_{vs_c}(T_{a_c})) = \frac{D_{v_c} \rho_{v_c}}{l \rho_{i_c}} \tag{19}$$

Dimensionless numbers $\left[ \mathrm{F}_i^T \right]$ and $\left[ \mathrm{F}_a^T \right]$ correspond to the inverse of the Fourier number in $\Omega_i$ and $\Omega_a$, respectively. They characterize the ratio between the rate of thermal energy storage and the heat conduction rate. $[\mathrm{F}_a^\rho]$ is an analogous inverse Fourier number for the transient water vapor transfer by diffusion in $\Omega_a$. Dimensionless numbers $[\mathrm{K}]$, $[\mathrm{R}]$, $[\mathrm{H}]$ and $[\mathrm{W_R}]$ are

defined at the ice-air interface. $[\mathrm{H}]$ characterizes the ratio between the heat flux induced by deposition and sublimation and the heat flux by conduction in the air phase. The above analysis slightly differs from the one presented in Calonne et al. (2014b). Indeed, two new dimensionless parameters are introduced: $[\mathrm{W_R}]$ and $[\mathrm{R}]$ to better capture the effect of $\alpha$ on the macroscopic models. Finally, let us remark that Eq. (18) defined at the ice-air interface corresponds to a Robin boundary condition, i.e a weighted combination of a Dirichlet boundary condition and a Neumann boundary condition. Hence, when $[\mathrm{W_R}]$ tends towards

zero, Eq. (18) is equivalent to a Neumann boundary condition ($D_v^* \mathbf{grad}^* \rho_v^* \cdot \mathbf{n_i} = 0$), whereas when $[\mathrm{W_R}]$ tends towards infinite (or is very large), Eq.(18) is equivalent to a Dirichlet boundary condition ($\rho_v^* = \rho_{vs}^*(T_a^*)$).

## 2.4   Estimation of the dimensionless numbers

The next key step is to estimate the above six dimensionless numbers with respect to the separation of scale parameter $\varepsilon = l/L$ in order to weigh the relative importance of the physical phenomena arising from the pore scale. In practice, $l$ and $L$ correspond

to the order of magnitude of the typical snow grain size and the thickness of a snow layer, respectively. In what follows, we assumed that $l \approx 5 \times 10^{-4}$ m and $L \approx 0.1$ m, leading to $\varepsilon = 5 \times 10^{-3}$. The characteristic value of each variable in the dimensionless numbers are summarized in Table 1. These values were evaluated for a temperature of -10°C and come from the literature (Massman, 1998; Kaempfer and Plapp, 2009). According to these characteristic values, it can be first shown (Calonne et al., 2014b) that the thermal diffusivity in the ice phase $D_{i_c} = k_{i_c}/(C_{i_c}\rho_{i_c})$ and in the air phase $D_{a_c} = k_{a_c}/(C_{a_c}\rho_{a_c})$, are

of the same order of magnitude than the vapor diffusion coefficient $D_{v_c}$. Thus, the characteristic time $t_c$ associated with these transfers through the snowpack are of the same order of magnitude: $t_c = \mathcal{O}(L^2/D_{i_c}) = \mathcal{O}(L^2/D_{a_c}) = \mathcal{O}(L^2/D_{v_c})$. Hence, from Eq. (19), we get $\left[ \mathrm{F}_i^T \right] = \mathcal{O}\left( \left[ \mathrm{F}_a^T \right] \right) = \mathcal{O}([\mathrm{F}_a^\rho]) = \mathcal{O}(\varepsilon^2)$. At the ice-pore interface, from Eq. (19), we have $[\mathrm{K}] = \mathcal{O}(1)$ and $[\mathrm{R}] = \mathcal{O}(1)$. The latter estimation implies that the supersaturation $\sigma = (\rho_v - \rho_{vs})/\rho_{vs}$ varies between -1 and 13, which is consistent with the range of values classically considered (Libbrecht and Rickerby, 2013). The dimensionless number $[\mathrm{W_R}]$

can be written:

$$[\mathrm{W_R}] = \frac{l \alpha_c w_{k_c}}{D_{v_c}} = \frac{l^2}{D_{v_c}} \frac{\alpha_c w_{k_c}}{l} = \frac{\tau_d}{\tau_{\mathrm{sub/dep}}}$$



**Table 1.** Characteristic values of the properties evaluated at -10°C from the literature (Massman, 1998; Kaempfer and Plapp, 2009).

| Symbol | Description | Value |
|--------|-------------|-------|
| $T_{i_c}, T_{a_c}$ | temperature of ice, air | 263 K |
| $k_{i_c}$ | heat conductivity of ice | 2.3 $\mathrm{W\,m^{-1}\,K^{-1}}$ |
| $k_{a_c}$ | heat conductivity of air | 0.024 $\mathrm{W\,m^{-1}\,K^{-1}}$ |
| $C_{i_c}$ | specific heat capacity of ice | 2000 $\mathrm{J\,kg^{-1}\,K^{-1}}$ |
| $C_{a_c}$ | specific heat capacity of air | 1005 $\mathrm{J\,kg^{-1}\,K^{-1}}$ |
| $L_{sg_c}$ | latent heat of sublimation of ice | $2.60 \times 10^9$ $\mathrm{J\,m^{-3}}$ |
| $D_{v_c}$ | water vapor diffusion coefficient in air | $2.036 \times 10^{-5}$ $\mathrm{m^2\,s^{-1}}$ |
| $\rho_{v_c}$ | water vapor density in air | 0.002 $\mathrm{kg\,m^{-3}}$ |
| $\rho_{i_c}$ | ice density | 917 $\mathrm{kg\,m^{-3}}$ |
| $\rho_{a_c}$ | air density | 1.335 kg $\mathrm{m^{-3}}$ |
| $l$ | microscopic length | $5 \times 10^{-4}$ m |
| $L$ | macroscopic length | 0.1 m |

where $\tau_d = l^2/D_{v_c}$ is the characteristic time associated to water vapor diffusion at the pore scale and $\tau_{\mathrm{sub/dep}} = l/(\alpha_c w_{k_c})$ is the characteristic time associated to the sublimation-deposition process. This result shows that this ratio can take different orders of magnitude depending on the value of $\alpha_c$. Using the characteristic values given in Table 1, this ratio is equal to 1 for

a particular value of $\alpha_c$, noted $\alpha_T = D_{v_c}/(lw_{k_c}) \approx 3 \times 10^{-4}$. This value decreases when the characteristic length $l$ increases, such as values range between $10^{-3}$ for small grains ($\sim$0.1 mm) and $10^{-5}$ for very large grains ($\sim$5 mm). It also depends on temperature but the influence is negligible in the -30 to 0°C range. The $\alpha_T$-value characterizes the transition between two mechanisms which drive the water vapor transfer at the pore scale. When $\tau_d \ll \tau_{\mathrm{sub/dep}}$, i.e for $\alpha_c \ll \alpha_T$, the water vapor flux is limited by sublimation-deposition processes. This case is also called the 'slow kinetics case' in Fourteau et al. (2021a).

When $\tau_d \gg \tau_{\mathrm{sub/dep}}$, i.e. for $\alpha_c \gg \alpha_T$, the water vapor transfer is mainly limited by diffusion, which is called 'fast kinetics case' in Fourteau et al. (2021a). For intermediate cases, both mechanisms may be in competition.

Estimations of the dimensionless numbers [H] is not as straightforward, as it depends on the intensity of the interface normal growth velocity $w_{n_c}$. When $\alpha_c$ is small (typically smaller than $\alpha_T$), $[W_R]$ is also small and Eq. (18) implies that $\Delta\rho_{v_c}$ has a finite value ($\mathcal{O}(\rho_{vc})$). Thus, this dimensionless number [H] can be also written:

$$[\mathrm{H}] = \frac{lL_{sg_c}\alpha_c w_{k_c}\rho_{v_c}}{k_{a_c}T_{a_c}\rho_{i_c}}.$$





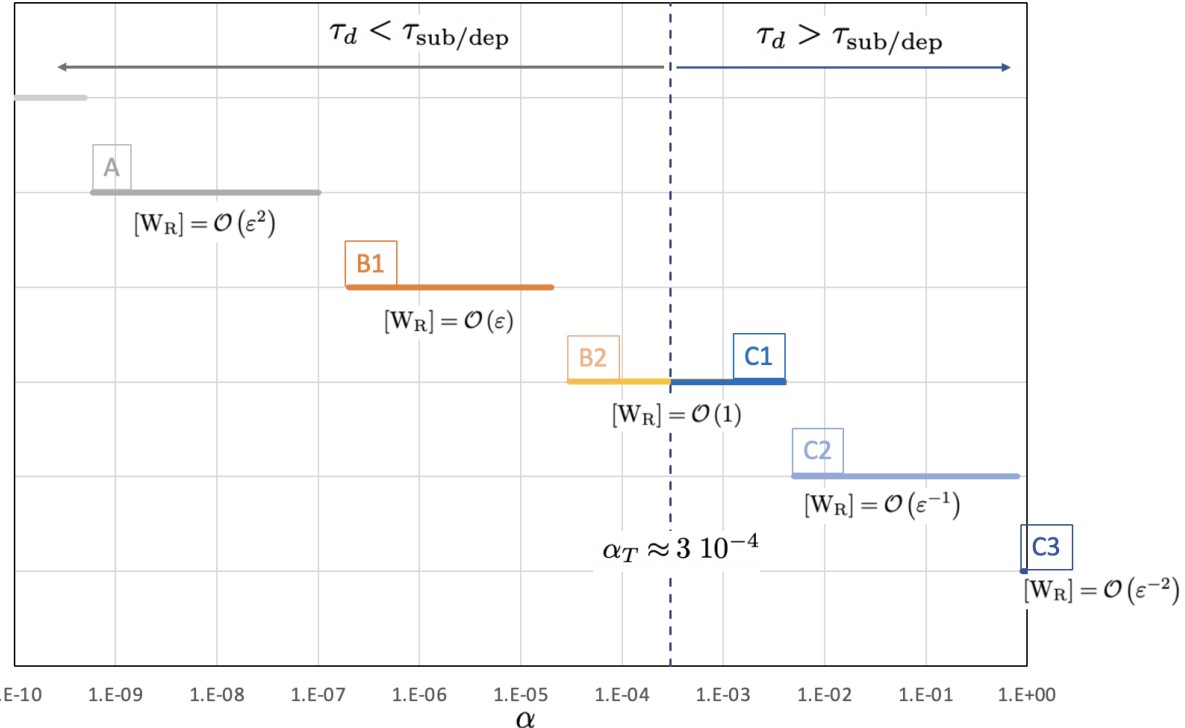

**Figure 2.** Estimation of the dimensionless number $[W_R]$ with respect to $\alpha$, which leads to several cases of macroscopic modeling to be considered (Cases A to C3). The $\alpha_T$-value characterizes the transition between two cases presenting different limiting processes for the water vapor transfer at the pore scale, so that $\tau_d < \tau_{\mathrm{sub/dep}}$ or $\tau_d > \tau_{\mathrm{sub/dep}}$, with $\tau_d$ the characteristic time associated to water vapor diffusion and $\tau_{\mathrm{sub/dep}}$ the characteristic time associated to the sublimation-deposition process. $\alpha_T$ was estimated based on the characteristic values given in Table 1.

In that case, it increases when $\alpha_c$ increases and according to the characteristic values given in Table 1, it is of the same order of $[W_R]$. For large values of $\alpha_c$ (typically larger than $\alpha_T$), Eq. (18) implies that $\rho_v^* \approx \rho_{vs}^*(T_a^*)$. As a consequence, from Eq. (17) and Eq. (19), $[H]$ can be rewritten:

$$[H] = \frac{l L_{sg_c} D_{vc} \gamma(T_{a_c}) T_{a_c}}{l \rho_{i_c} k_{a_c} T_{a_c}} = \frac{L_{sg_c} D_{vc} \gamma(T_{a_c})}{\rho_{i_c} k_{a_c}} = \frac{k_{\mathrm{dif}_c}}{k_{a_c}}$$

where $\gamma(T_{a_c}) = \mathrm{d}\rho_{vs}(T_{a_c})/\mathrm{d}T_{a_c}$ is the derivative of Clausius-Clapeyron's law and $k_{\mathrm{dif}_c} = L_{sg_c} D_{vc} \gamma(T_{a_c})/\rho_{i_c}$ can be seen as 'an enhancement' of the air thermal conductivity. Using the characteristic values given in Table 1 and the Clausius-Clapeyron Eq. (9), for large values of $\alpha_c$, $[H] = \mathcal{O}(1)$. According to the above analysis, several cases must be considered depending on the value of the condensation coefficient $\alpha_c$ (Fig. 2):

– Case A: $\tau_d = \mathcal{O}(\varepsilon^2 \tau_{\mathrm{sub/dep}})$, i.e $[W_R] = \mathcal{O}(\varepsilon^2)$ and $[H] = \mathcal{O}(\varepsilon^2)$

– Case B1: $\tau_d = \mathcal{O}(\varepsilon \tau_{\mathrm{sub/dep}})$, i.e $[W_R] = \mathcal{O}(\varepsilon)$ and $[H] = \mathcal{O}(\varepsilon)$





– Case B2: $\tau_d = \mathcal{O}(\tau_{\mathrm{sub/dep}})$, i.e $[\mathrm{W_R}] = \mathcal{O}(1)$ but with $\varepsilon^{1/2} \leqslant [\mathrm{W_R}] \leqslant 1$ and $[\mathrm{H}] = \mathcal{O}(1)$

– Case C1: $\tau_d = \mathcal{O}(\tau_{\mathrm{sub/dep}})$, i.e $[\mathrm{W_R}] = \mathcal{O}(1)$ but with $1 \leqslant [\mathrm{W_R}] \leqslant \varepsilon^{-1/2}$ and $[\mathrm{H}] = \mathcal{O}(1)$

– Case C2: $\tau_d = \mathcal{O}(\varepsilon^{-1}\tau_{\mathrm{sub/dep}})$, i.e $[\mathrm{W_R}] = \mathcal{O}\left(\varepsilon^{-1}\right)$ and $[\mathrm{H}] = \mathcal{O}(1)$

– Case C3: $\tau_d = \mathcal{O}(\varepsilon^{-2}\tau_{\mathrm{sub/dep}})$, i.e $[\mathrm{W_R}] = \mathcal{O}\left(\varepsilon^{-2}\right)$ and $[\mathrm{H}] = \mathcal{O}(1)$

The cases A, B1 and B2 correspond to $0 \leqslant \alpha \leqslant \alpha_T$, whereas the cases C1, C2 and C3 correspond to $\alpha_T \leqslant \alpha \leqslant 1$. Moreover, let us remark that the cases B2 and C1 correspond to the same order of magnitude of $[\mathrm{W_R}]$ and $[\mathrm{H}]$. However, two cases are considered to take into account the transition which occurs when $\alpha = \alpha_T$ (see Eq. (18)).

## 2.5   Asymptotic analysis

   The next step is to introduce multiple-scale coordinates (Bensoussan et al., 1978; Sanchez-Palencia, 1980; Auriault, 1991).
The two characteristic lengths $L$ and $l$ introduce two dimensionless space variables, $\mathbf{x}^* = \mathbf{X}/L$ and $\mathbf{y}^* = \mathbf{X}/l$, where $\mathbf{X}$ is the physical space variable. The macroscopic (or slow) dimensionless space variable $\mathbf{x}^*$ is related to the microscopic (or fast) dimensionless space variable $\mathbf{y}^*$ by $\mathbf{x}^* = \varepsilon\mathbf{y}^*$. When $l$ is used as the characteristic length, the dimensionless derivative operator $\mathbf{grad}^*$ becomes $(\mathbf{grad}_{y^*} + \varepsilon\,\mathbf{grad}_{x^*})$, where the subscripts $_{x^*}$ and $_{y^*}$ denote the derivatives with respect to the variables $\mathbf{x}^*$ and $\mathbf{y}^*$, respectively. Following the multiple-scale expansion technique (Bensoussan et al., 1978; Sanchez-Palencia, 1980;
Auriault, 1991), the ice temperature $T_i^*$, the air temperature $T_a^*$, and the water vapor $\rho_v^*$ are sought in the form of asymptotic expansions of powers of $\varepsilon$:

$$\varphi^*(\mathbf{x}^*,\mathbf{y}^*,t) = \varphi^{*(0)}(\mathbf{x}^*,\mathbf{y}^*,t) + \varepsilon\varphi^{*(1)}(\mathbf{x}^*,\mathbf{y}^*,t) + \varepsilon^2\varphi^{*(2)}(\mathbf{x}^*,\mathbf{y}^*,t) + ... \tag{20}$$

where $\varphi^* = T_i^*, T_a^*, \rho_v^*$ and the corresponding $\varphi^{*(i)}$ are periodic functions of period $\Omega$ with respect to the space variable $\mathbf{y}^*$. Substituting these expansions in the set (13)-(18) gives, by identification of like powers of $\varepsilon$, successive boundary value
problems to be investigated. All the details concerning this asymptotic analysis are presented in the Supplement. The main results are summarized in the following section.

## 2.6   Macroscopic equivalent descriptions

### 2.6.1   Case A

   The case A corresponds to the model presented in Calonne et al. (2014b). According to the order of magnitude of the dimen-
sionless numbers and notably $[\mathrm{H}] = \mathcal{O}\left(\varepsilon^2\right), [\mathrm{W_R}] = \mathcal{O}\left(\varepsilon^2\right)$, the asymptotic analysis presented in the Supplement (Sec. S1) shows that the heat transfer and the water vapor diffusion at the macroscopic scale are described by the equations (A.45) and (A.48). Returning in dimensional variables, the macroscopic model is written:

$$(\rho C)^{\mathrm{eff}}\frac{\partial T^{(0)}}{\partial t} - \mathrm{div}(\mathbf{k}^{\mathrm{eff}}\mathbf{grad}\,T^{(0)}) = \mathrm{SSA_V}L_{sg}w_n^{(0)} = -L_{sg}\dot{\phi} \tag{21}$$



$$\phi \frac{\partial \rho_v^{(0)}}{\partial t} - \mathrm{div}(\mathbf{D}^{\mathrm{eff}}\mathbf{grad}\,\rho_v^{(0)}) = -\mathrm{SSA_V}\rho_i w_n^{(0)} = \rho_i \dot{\phi} \tag{22}$$

where $w_n^{(0)}$ is given by the Hertz-Knudsen Eq. (A.44) and the Clausius-Clapeyron's law (A.43)

$$w_n^{(0)} = \frac{\alpha}{\rho_i} w_{\mathrm{k}} \left[ \rho_v^{(0)} - \rho_{vs}^{(0)}(T^{(0)}) \right] \tag{23}$$

$$\rho_{vs}^{(0)}(T^{(0)}) = \rho_{vs}^{\mathrm{ref}} \exp\left[ \frac{L_{sg}m}{\rho_i k}\left( \frac{1}{T^{\mathrm{ref}}} - \frac{1}{T^{(0)}} \right) \right] \tag{24}$$

and where $\phi$ is the porosity and $\dot{\phi}$ its total time derivative. $\mathrm{SSA_V} = |\Gamma|/|\Omega|$ is the specific surface area per unit volume, defined as the ice surface area over the snow volume in m$^{-1}$. The SSA can also be defined per unit mass, with $\mathrm{SSA_V} = \mathrm{SSA} \times \rho_i$. $(\rho C)^{\mathrm{eff}}$ is the effective thermal capacity (A.46), $k^{\mathrm{eff}}$ is the effective thermal conductivity tensor (A.47), and $\mathbf{D}^{\mathrm{eff}}$ is the effective diffusion tensor (A.49). These effective properties are defined as:

$$(\rho C)^{\mathrm{eff}} = (1 - \phi)\rho_i C_i + \phi \rho_a C_a \tag{25}$$

$$\mathbf{k}^{\mathrm{eff}} = \frac{1}{|\Omega|}\left( \int_{\Omega_a} k_a(\mathbf{grad}\,\mathbf{t}_a + \mathbf{I})\mathrm{d}\Omega + \int_{\Omega_i} k_i(\mathbf{grad}\,\mathbf{t}_i + \mathbf{I})\mathrm{d}\Omega \right) \tag{26}$$

$$\mathbf{D}^{\mathrm{eff}} = \frac{1}{|\Omega|}\int_{\Omega_a} D_v(\mathbf{grad}\,\mathbf{g}_v + \mathbf{I})\mathrm{d}\Omega \tag{27}$$

where $\mathbf{t}_a$ and $\mathbf{t}_i$ are two periodic vectors, solution of the following boundary value problem over the REV (A.20)-(A.24):

$$\mathrm{div}(k_i(\mathbf{grad}\,\mathbf{t}_i + \mathbf{I})) = 0 \quad \text{in } \Omega_i \tag{28}$$

$$\mathrm{div}(k_a(\mathbf{grad}\,\mathbf{t}_a + \mathbf{I})) = 0 \quad \text{in } \Omega_a \tag{29}$$

$$\mathbf{t}_i = \mathbf{t}_a \quad \text{on } \Gamma \tag{30}$$

$$(k_i(\mathbf{grad}\,\mathbf{t}_i + \mathbf{I}) - k_a(\mathbf{grad}\,\mathbf{t}_a + \mathbf{I})) \cdot \mathbf{n_i} = 0 \quad \text{on } \Gamma \tag{31}$$

$$\frac{1}{|\Omega|}\int_{\Omega}(\mathbf{t_a} + \mathbf{t_i})\mathrm{d}\Omega = \mathbf{0} \tag{32}$$

and where $\mathbf{g}_v$ is a periodic vector solution of the following boundary value problem over the REV (A.36)-(A.38):

$$\mathrm{div}(D_v(\mathbf{grad}\,\mathbf{g}_v + \mathbf{I})) = 0 \quad \text{in } \Omega_a \tag{33}$$

$$D_v(\mathbf{grad}\,\mathbf{g}_v + \mathbf{I}) \cdot \mathbf{n_i} = 0 \quad \text{on } \Gamma \tag{34}$$





$$\frac{1}{|\Omega|} \int_{\Omega_a} \mathbf{g}_v \, \mathrm{d}\Omega = \mathbf{0} \tag{35}$$

In that case, the above macroscopic equivalent description shows that, at the first order, the heat and water vapor transfer are described by two equations which are coupled through a source term proportional to the Hertz-Knudsen equation (23) and the Clausius Clapeyron's law (24), but expressed with respect to the two macroscopic variables $T^{(0)}$ and $\rho_v^{(0)}$. These equations involve two effective parameters: the effective thermal conductivity $\mathbf{k}^{\mathrm{eff}} = \mathbf{k}^{\mathrm{eff}}(k_i, k_a, \text{microstructure})$ and the effective diffusion $\mathbf{D}^{\mathrm{eff}} = \mathbf{D}^{\mathrm{eff}}(D_v, \text{microstructure})$.

### 2.6.2 Cases B1 and B2

According to the order of magnitude of the dimensionless numbers in the cases B1 and B2, the asymptotic analysis presented in the Supplement (Sec. S2) shows that these two cases lead to the same macroscopic description: the heat transfer and the water vapor diffusion at the macroscopic scale are described by the equations (B1.29) and (B1.44) (or (B2.41) and (B2.46)). Returning in dimensional variables, the macroscopic model is written:

$$(\rho C)^{\mathrm{eff}} \frac{\partial T^{(0)}}{\partial t} - \mathrm{div}(\mathbf{k}^{\mathrm{eff}} \mathbf{grad} T^{(0)}) = -L_{sg} \dot{\phi} \tag{36}$$

$$\phi \frac{\partial \rho_{vs}^{(0)}}{\partial t} - \mathrm{div}(\mathbf{D}^{\mathrm{eff}} \mathbf{grad} \rho_{vs}^{(0)}) = \rho_i \dot{\phi} \tag{37}$$

with

$$\rho_v^{(0)} = \rho_{vs}^{(0)}(T^{(0)}) \tag{38}$$

where $(\rho C)^{\mathrm{eff}}$ is the effective thermal capacity, $\mathbf{k}^{\mathrm{eff}}$ is the effective thermal conductivity tensor and $\mathbf{D}^{\mathrm{eff}}$ is the effective diffusion tensor as defined in the case A. The above macroscopic equivalent description shows that at the first order the heat and vapor transfer are only driven by the temperature field, since the water vapor density $\rho_v^{(0)} = \rho_{vs}^{(0)}(T^{(0)})$ is directly given by the Clausius-Clapeyron equation (24). Consequently, from (37) we have:

$$\dot{\phi} = -\frac{1}{\rho_i} \left( \mathrm{div}(\mathbf{D}^{\mathrm{eff}} \mathbf{grad} \rho_{vs}^{(0)}(T^{(0)})) - \phi \frac{\partial \rho_{vs}^{(0)}}{\partial t} \right) = -\frac{1}{\rho_i} \left( \mathrm{div}(\gamma(T^{(0)}) \mathbf{D}^{\mathrm{eff}} \mathbf{grad} T^{(0)}) - \phi \gamma(T^{(0)}) \frac{\partial T^{(0)}}{\partial t} \right) \tag{39}$$

where

$$\gamma(T^{(0)}) = \frac{\mathrm{d}\rho_{vs}^{(0)}(T^{(0)})}{\mathrm{d}T^{(0)}} = \rho_{vs}^{\mathrm{ref}} \frac{L_{sg} m}{\rho_i k} \frac{1}{(T^{(0)})^2} \exp\left[ \frac{L_{sg} m}{\rho_i k} \left( \frac{1}{T^{\mathrm{ref}}} - \frac{1}{T^{(0)}} \right) \right] = \frac{L_{sg} m}{\rho_i k} \frac{1}{(T^{(0)})^2} \rho_{vs}^{(0)}(T^{(0)}) \tag{40}$$

Taking into account this result, the macroscopic heat transfer equation (36) is written:

$$\left( (\rho C)^{\mathrm{eff}} + \phi \gamma(T^{(0)}) \frac{L_{sg}}{\rho_i} \right) \frac{\partial T^{(0)}}{\partial t} - \mathrm{div}(\mathbf{k}^{\mathrm{B}} \mathbf{grad} T^{(0)}) = 0 \tag{41}$$



In this latter equation,

$$\mathbf{k}^{\mathrm{B}} = \mathbf{k}^{\mathrm{eff}} + \frac{\gamma(T^{(0)})L_{sg}}{\rho_i}\mathbf{D}^{\mathrm{eff}} \tag{42}$$

appears as an apparent thermal conductivity of the snow which depends non-linearly on the temperature through $\gamma(T^{(0)})$. Our results show that this is valid if $[\mathrm{W_R}] = \mathcal{O}(\varepsilon)$ or $\mathcal{O}(1)$, i.e for $\alpha$-values ranging from around $10^{-6}$ to $\alpha_T$, typically. Finally, let us remark that (i) this model B can be also seen as a particular case of the model A, when $\rho_v^{(0)}$ tends towards $\rho_{vs}^{(0)}(T^{(0)})$ by increasing $\alpha$, and (ii) the apparent thermal conductivity of the snow $\mathbf{k}^{\mathrm{B}}$ can be also written:

$$\mathbf{k}^{\mathrm{B}} = \mathbf{k}^{\mathrm{eff}} + \frac{\gamma(T^{(0)})L_{sg}D_v}{\rho_i}\frac{\mathbf{D}^{\mathrm{eff}}}{D_v} = \mathbf{k}^{\mathrm{eff}} + k_{\mathrm{dif}}\frac{\mathbf{D}^{\mathrm{eff}}}{D_v} \tag{43}$$

where $k_{\mathrm{dif}} = \gamma(T^{(0)})L_{sg}D_v/\rho_i$ corresponds to "an enhancement" of the air thermal conductivity, as defined in Sect. 2.4. However, in that case, $\gamma(T^{(0)})$ depends on the macroscopic temperature $T^{(0)}$.

### 2.6.3 Cases C1, C2 and C3

According to the order of magnitude of the dimensionless numbers in the cases C1, C2 and C3, the asymptotic analysis presented in the Supplement (Sec. S3) shows that these two cases lead to the same macroscopic description. Returning in dimensional variables, the macroscopic model (C1.19-C1.22) (see also (C2.41-C2.44) and (C3.41-C3.44)) is written:

$$(\rho C)^{\mathrm{eff}}\frac{\partial T^{(0)}}{\partial t} - \mathrm{div}(\mathbf{k}^{\mathrm{td}}\mathbf{grad}T^{(0)}) = -L_{sg}\dot{\phi} \tag{44}$$

$$\phi\frac{\partial \rho_{vs}^{(0)}}{\partial t} - \mathrm{div}(\mathbf{D}^{\mathrm{td}}\mathbf{grad}\rho_{vs}^{(0)}) = \rho_i\dot{\phi} \tag{45}$$

$$\rho_v^{(0)} = \rho_{vs}^{(0)}(T^{(0)}) \tag{46}$$

$(\rho C)^{\mathrm{eff}}$ is the classical dimensionless effective thermal capacity. The macroscopic thermal conductivity tensor $\mathbf{k}^{\mathrm{td}}$ and the macroscopic diffusion tensor $\mathbf{D}^{\mathrm{td}}$ are defined as

$$\mathbf{k}^{\mathrm{td}} = \frac{1}{|\Omega|}\left(\int_{\Omega_a} k_a(\mathbf{grad}\,\mathbf{r}_a + \mathbf{I})\mathrm{d}\Omega + \int_{\Omega_i} k_i(\mathbf{grad}\,\mathbf{r}_i + \mathbf{I})\mathrm{d}\Omega\right) \tag{47}$$

$$\mathbf{D}^{\mathrm{td}} = \frac{1}{|\Omega|}\int_{\Omega_a} D_v(\mathbf{grad}\,\mathbf{r}_a + \mathbf{I})\mathrm{d}\,\Omega \tag{48}$$

where $\mathbf{r}_a$ and $\mathbf{r}_i$ are two periodic vectors, solution of the following boundary value problem over the REV (C1.8)-(C1.12):

$$\mathrm{div}(k_i(\mathbf{grad}\,\mathbf{r}_i + \mathbf{I})) = 0 \quad \mathrm{in}\ \Omega_i \tag{49}$$

$$\mathrm{div}((k_a + k_{\mathrm{dif}})(\mathbf{grad}\,\mathbf{r}_a + \mathbf{I})) = 0 \quad \mathrm{in}\ \Omega_a \tag{50}$$





$$\mathbf{r}_i = \mathbf{r}_a \quad \text{on } \Gamma \tag{51}$$

$$(k_i(\mathbf{grad\ r}_i + \mathbf{I}) - (k_a + k_{\text{dif}})(\mathbf{grad\ r}_a + \mathbf{I})) \cdot \mathbf{n_i} = 0 \quad \text{on } \Gamma \tag{52}$$


$$\frac{1}{|\Omega|} \int_\Omega (\mathbf{r}_a + \mathbf{r}_i)\mathrm{d}\Omega = \mathbf{0} \tag{53}$$

As for the model B, the macroscopic heat transfer equation (44) can be also written:

$$\left((\rho C)^{\text{eff}} + \phi\gamma(T^{(0)})\frac{L_{sg}}{\rho_i}\right)\frac{\partial T^{(0)}}{\partial t} - \text{div}(\mathbf{k}^{\text{C}}\mathbf{grad}T^{(0)}) = 0 \tag{54}$$

In this latter equation,

$$\mathbf{k}^{\text{C}} = \mathbf{k}^{\text{td}} + \frac{\gamma(T^{(0)})L_{sg}}{\rho_i}\mathbf{D}^{\text{td}} = \mathbf{k}^{\text{td}} + k_{\text{dif}}\frac{\mathbf{D}^{\text{td}}}{D_v} \tag{55}$$

appears as an apparent thermal conductivity of the snow. In that case $\mathbf{k}^{\text{td}}$ and $\mathbf{D}^{\text{td}}$ both depend on $k_a$, $k_i$, and $k_{\text{dif}}$ and we have:

$$\mathbf{k}^{\text{C}} = \frac{1}{|\Omega|}\left(\int_{\Omega_a}(k_a + k_{\text{dif}})(\mathbf{grad\ r}_a + \mathbf{I})\mathrm{d}\Omega + \int_{\Omega_i}k_i(\mathbf{grad\ r}_i + \mathbf{I})\mathrm{d}\Omega\right) \tag{56}$$

This model C corresponds to the one derived by Moyne et al. (1988), assuming that $\rho_v = \rho_{vs}(T)$ on the interface at the
microscopic scale and using the volume averaging-method. This model is also similar to the one derived by Hansen and
Foslien (2015), assuming that $\alpha \approx 10^{-2}$. In that case, we show that this model is valid for $\alpha$-values ranging from $\alpha_T$ to 1.

## 2.7 Macroscopic equivalent descriptions - synthesis

Figure 3 presents a summary of the three macroscopic models of heat and vapor transport in dry snow derived above, together
with their domain of validity according to the value of $\alpha$. As already mentioned, the model A is the one already derived in
Calonne et al. (2014b), whereas the model C is equivalent to the model derived by Moyne et al. (1988) and Hansen and Foslien
(2015). In practice, the model A and C are sufficient to describe the heat and vapor transfer in the whole range of $\alpha$, since the
model B can be seen as a particular case of the model A, assuming that $\rho_v$ tends towards $\rho_{vs}(T)$ at macro-scale. A transition
value $\alpha_T$ taken at $3 \times 10^{-4}$ was presented, yet we recall that it can vary between $10^{-5}$ and $10^{-3}$ depending on the grain size
and, to a lesser extend, temperature. Besides, without loss of generality, a constant value of $\alpha$ accounting for sublimation and
deposition (in time, space and other dependencies) was used, material properties such as $D_v$, $k_i$ and $k_a$ were taken constant at
-10°C, and curvature effects were neglected.

The hypothesis that $\rho_v = \rho_{vs}(T)$, which is often made, appears as a good approximation for $\alpha$-values larger than $10^{-6}$.
However, the asymptotic analysis shows that in the range $[10^{-6}, \alpha_T]$, this approximation is of the order of $\mathcal{O}(\varepsilon)$ since
$\rho_v^{(0)} = \rho_{vs}^{(0)}(T^{(0)})$, i.e. $\sigma = (\rho_v - \rho_{vs})/\rho_{vs} \approx \mathcal{O}(\varepsilon)$. In the range $[\alpha_T, 1]$, this approximation is of the order of $\mathcal{O}(\varepsilon^2)$, since





**Figure 3.** Definition of the three different macroscopic models and their domain of validity with respect to $\alpha$. The value of $\alpha_T$ was estimated based on the characteristic values given in Table 1.

$\rho_v^{(0)} = \rho_{vs}^{(0)}\left(T^{(0)}\right)$ and $\rho_v^{(1)} = \rho_{vs}^{(1)}\left(T^{(1)}\right)$, i.e. $\sigma \approx \mathcal{O}\left(\varepsilon^2\right)$. It is also worth noting that the value of $\alpha$ appears explicitly in the model A only.

In the model A and B, the water vapor transfer are mainly limited by the sublimation-deposition at the ice-air interfaces. At macro-scale, diffusion is characterized by the classical effective diffusion $\mathbf{D}^{\text{eff}}(D_v, \text{microstructure})$ which depends on $D_v$ and the microstructure of the snow. In the model C, the water vapor transfer is mainly limited by the diffusion process at micro-scale. In that case, the macroscopic diffusion tensor $\mathbf{D}^{\text{td}}(D_v, k_i, k_a, k_{\text{dif}}, \text{microstructure})$ appears as a "thermo-diffusion" coefficient since it depends on $D_v$, the microstructure of the snow, but also on the thermal properties of the ice $k_i$, and the air $k_a + k_{\text{dif}}$, the latter being enhanced by the phase change through $k_{\text{dif}}$.

Even if the model B and C can be written in a similar form, the involved macroscopic parameters strongly differ, since they capture different mechanisms (diffusion or sublimation-deposition) arising at the pore scale, and consequently they cannot be deduced from one model to the other in a simple way.





# 3   Application to analytical and numerical cases

In this section, two simple snow microstructures, a bilayer and an assemblage of spherical grains and pores, are first considered to illustrate the influence of the microstructure and of the parameters taken at the pore scale on the macroscopic parameters of models A, B and C (Sec. 3.2 and 3.1).

Then, a simplified 2D snow microstructure is considered to evaluate the models by comparing simulation results obtained with the pore scale description and with the macroscopic modelings (Sec. 3.3).

## 3.1   The bilayer snowpack: upper and lower bounds

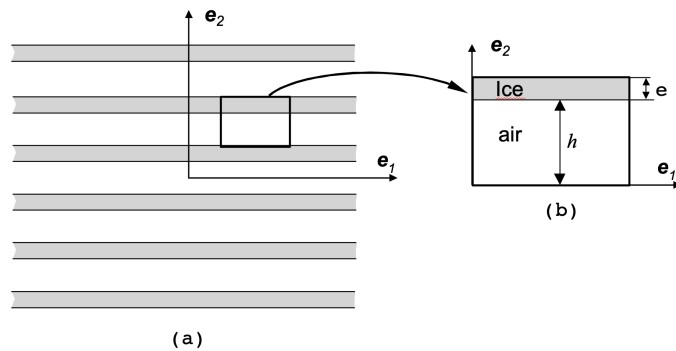

**Figure 4.** Illustration of the bilayer snowpack problem: (a) at the macroscopic scale, (b) at the scale of a Representative Elementary Volume (REV).

As a first example, we consider the classical bilayer material problem and the snowpack is seen as a succession of horizontal layers of pure air and of pure ice, as illustrated in Fig. 4. In this case, the macroscopic parameters arising in the models A, B

and C can be analytically determined and constitute the upper and lower bounds of these parameters for any anisotropic snow microstructure. The boundary value problems (33-35), (28-32), (49-53) have been solved analytically on the REV (Fig. 4.b) in Auriault et al. (2009). Taking into account those results and using equations (27) and (26), we have for the model A and B:

$$\mathbf{D}^{\text{eff}} = \begin{pmatrix} D_{11}^{\text{eff}} & 0 \\ 0 & 0 \end{pmatrix} \quad D_{11}^{\text{eff}} = \phi D_v \tag{57}$$

$$\mathbf{k}^{\text{eff}} = \begin{pmatrix} k_{11}^{\text{eff}} & 0 \\ 0 & k_{22}^{\text{eff}} \end{pmatrix} \quad k_{11}^{\text{eff}} = \phi k_a + (1-\phi)k_i, \quad k_{22}^{\text{eff}} = \frac{k_i k_a}{(1-\phi)k_a + \phi k_i} \tag{58}$$

Thus, it comes that;

$$k_{11}^{\text{B}} = \phi(k_a + k_{\text{dif}}) + (1-\phi)k_i, \quad k_{22}^{\text{B}} = \frac{k_i k_a}{(1-\phi)k_a + \phi k_i} \tag{59}$$

These results imply that the macroscopic properties $(D^{\text{eff}}, k^{\text{eff}}, k^{\text{B}})$ of any anisotropic snow verify the following bounds:

$$0 \leqslant D^{\text{eff}} \leqslant \phi D_v, \tag{60}$$




and,

$$\frac{k_i k_a}{(1-\phi)k_a + \phi k_i} \leqslant k^{\mathrm{eff}} \leqslant \phi k_a + (1-\phi)k_i, \quad \frac{k_i k_a}{(1-\phi)k_a + \phi k_i} \leqslant k^{\mathrm{B}} \leqslant \phi(k_a + k_{\mathrm{dif}}) + (1-\phi)k_i \tag{61}$$

For the model C, from (48) and (47), we have:

$$\mathbf{D}^{\mathrm{td}} = \begin{pmatrix} D_{11}^{\mathrm{td}} & 0 \\ 0 & D_{22}^{\mathrm{td}} \end{pmatrix} \quad D_{11}^{\mathrm{td}} = \phi D_v, \quad D_{22}^{\mathrm{td}} = \phi D_v \frac{k_i}{(1-\phi)(k_a + k_{\mathrm{dif}}) + \phi k_i} \tag{62}$$

$\quad \mathbf{k}^{\mathrm{td}} = \begin{pmatrix} k_{11}^{\mathrm{td}} & 0 \\ 0 & k_{22}^{\mathrm{td}} \end{pmatrix} \quad k_{11}^{\mathrm{td}} = \phi k_a + (1-\phi)k_i, \quad k_{22}^{\mathrm{td}} = \frac{k_i(k_a + (1-\phi)k_{\mathrm{dif}})}{(1-\phi)(k_a + k_{\mathrm{dif}}) + \phi k_i} \tag{63}$

Thus, it comes that

$$k_{11}^{\mathrm{C}} = \phi(k_a + k_{\mathrm{dif}}) + (1-\phi)k_i, \quad k_{22}^{\mathrm{C}} = \frac{k_i(k_a + k_{\mathrm{dif}})}{(1-\phi)(k_a + k_{\mathrm{dif}}) + \phi k_i} \tag{64}$$

In that case, these results imply that the macroscopic properties $(D^{\mathrm{td}}, k^{\mathrm{td}}, k^{\mathrm{C}})$ of any anisotropic snow verify the following bounds:

$\quad \phi D_v \leqslant D^{\mathrm{td}} \leqslant \phi D_v \dfrac{k_i}{(1-\phi)(k_a + k_{\mathrm{dif}}) + \phi k_i} \tag{65}$

and

$$\frac{k_i(k_a + (1-\phi)k_{\mathrm{dif}})}{(1-\phi)(k_a + k_{\mathrm{dif}}) + \phi k_i} \leqslant k^{\mathrm{td}} \leqslant \phi k_a + (1-\phi)k_i, \quad \frac{k_i(k_a + k_{\mathrm{dif}})}{(1-\phi)(k_a + k_{\mathrm{dif}}) + \phi k_i} \leqslant k^{\mathrm{C}} \leqslant \phi(k_a + k_{\mathrm{dif}}) + (1-\phi)k_i \tag{66}$$

The above results show that, as already underlined in Calonne et al. (2014b), Moyne et al. (1988) and Fourteau et al. (2021b), the bounds (60) and (65) of both the effective diffusion coefficients $D^{\mathrm{eff}}$ and $D^{\mathrm{td}}$ are always smaller than $D_v$ and $D^{\mathrm{td}} > D^{\mathrm{eff}}$,

whatsoever the $\alpha$-value. Moreover, according to the definition of $\mathbf{D}^{\mathrm{eff}}$ (Eq. 57) and $\mathbf{D}^{\mathrm{td}}$ (Eq. 62), if a vertical macroscopic temperature gradient is applied along $\mathbf{e}_2$, the model A (or B) will not predict any mass variation along that direction because of the pore geometry. By contrast, the model C, where the sublimation-deposition process is faster than diffusion, can predict mass transport along $\mathbf{e}_2$ since $D_{22}^{\mathrm{td}} \neq 0$.

### 3.2 Assemblage of spherical grains and pores: self-consistent estimates

The next analytical model is the self-consistent model (Bruggeman, 1935; Hill, 1965; Budiansky, 1965; Torquato, 2002). Previous works showed that self-consistent (SC) estimates provide good estimations of the macroscopic properties of heat and vapor transport in dry snow (Calonne et al., 2014b, a, 2019). In this model, the snow microstructure is considered as a macroscopically isotropic material made of an assemblage of spherical inclusions of air or ice. Each type of inclusion is embedded in a homogeneous equivalent material, which allows accounting for the connectivity of both phases. The equivalent

material corresponds to an infinite matrix whose effective properties is the unknown to be calculated. The solution of the equations for an isolated inclusion then gives an implicit relation which can be solved for this effective property.







**Figure 5.** Evolution of the SC estimates of the thermal conductivities $k_{SC}^{eff}$, $k_{SC}^{B}$ and $k_{SC}^{C}$ with respect to porosity at four temperatures (a, c, e), and with respect to temperature for four porosities (b, d, f). The vertical dotted gray lines indicate the four temperature and porosity values considered.

For the model A, the SC estimate of the effective thermal conductivity of snow $k_{SC}^{eff}$ and of the effective diffusion coefficient $D_{SC}^{eff}$ verify the following implicit relation (Torquato, 2002):

$$k_{SC}^{eff} = \frac{\beta + \sqrt{\beta^2 + 8k_i k_a}}{4} \quad \text{with} \quad \beta = k_i(3(1-\phi) - 1) + k_a(3\phi - 1) \tag{67}$$






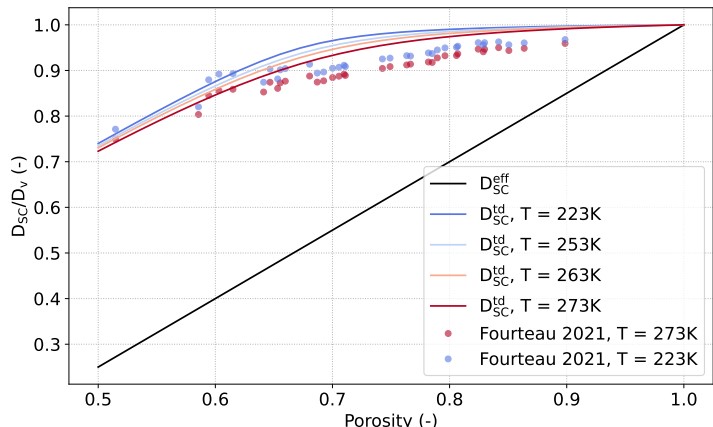

**Figure 6.** Evolution of the normalized SC estimates $D_{\mathrm{SC}}^{\mathrm{eff}}/D_v$ and $D_{\mathrm{SC}}^{\mathrm{td}}/D_v$ with respect to porosity at different temperatures (solid lines). Results of the numerical computations of $D_{\mathrm{SC}}^{\mathrm{td}}/D_v$ at two temperatures from Fourteau et al. (2021a) are also shown (symbols).

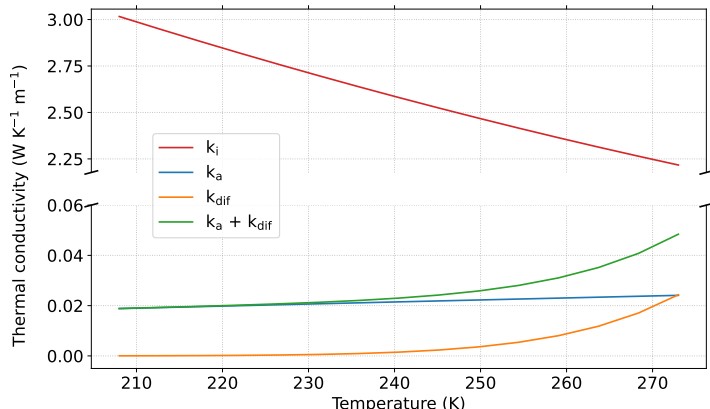

**Figure 7.** Evolution of the thermal conductivity with temperature for $k_i$ (Huang et al., 2013), $k_a$ (Haynes, 2016), $k_{\mathrm{dif}}$, and $k_a + k_{\mathrm{dif}}$.

$$D_{\mathrm{SC}}^{\mathrm{eff}} = D_v \frac{(3\phi - 1)}{2} \tag{68}$$

For the model B, the SC estimates of thermal conductivity $k_{\mathrm{SC}}^{\mathrm{B}}$ is simply obtained by replacing the effective properties by their SC estimates in Eq. (41) and reads:

$$k_{\mathrm{SC}}^{\mathrm{B}} = k_{\mathrm{SC}}^{\mathrm{eff}} + k_{\mathrm{dif}} \frac{D_{\mathrm{SC}}^{\mathrm{eff}}}{D_v} \tag{69}$$

Finally, for the model C, the SC estimate of thermal conductivity $k_{\mathrm{SC}}^{\mathrm{C}}$ can be obtained by replacing $k_a$ in Eq. (67) by $k_a + k_{\mathrm{dif}}$ as:

$$k_{\mathrm{SC}}^{\mathrm{C}} = \frac{\beta + \sqrt{\beta^2 + 8 k_i (k_a + k_{\mathrm{dif}})}}{4} \quad \text{with} \quad \beta = k_i(3(1 - \phi) - 1) + (k_a + k_{\mathrm{dif}})(3\phi - 1) \tag{70}$$



For the diffusion coefficient $D_{\text{SC}}^{\text{td}}$, it can be shown (Auriault et al., 2009) that:

$$D_{\text{SC}}^{\text{td}} = \phi D_v \frac{3k_{\text{SC}}^{\text{C}}}{(k_a + k_{\text{dif}}) + 2k_{\text{SC}}^{\text{C}}} \qquad (71)$$

The above SC estimates of thermal conductivity and diffusion coefficient are presented in Figure 5 and 6 and the impact of snow porosity and temperature is shown. To do so, we used the relationships of the thermal conductivity of ice $k_i(T)$ and of air $k_a(T)$ with temperature from Huang et al. (2013) and Haynes (2016), respectively.

For thermal conductivity, the SC estimates $k_{\text{SC}}^{\text{eff}}$, $k_{\text{SC}}^{\text{B}}$, and $k_{\text{SC}}^{\text{C}}$ at a given temperature are similar and follow the classical exponential evolution with snow porosity. Overall, estimates vary between about 0.06 W m$^{-1}$ K$^{-1}$ for porosity of 0.5 and 0.01

W m$^{-1}$ K$^{-1}$ for porosity of 1 (Fig. 5.a, 5.c and 5.e). More differences between the estimates can be seen for the normalized diffusion coefficient. The effective coefficient $D_{\text{SC}}^{\text{eff}}/D_v$ is overall much smaller than $D_{\text{SC}}^{\text{td}}/D_v$ and evolves linearly from 0.25 to 1 when porosity varies from 0.5 to 1 (Fig. 6). In contrast, $D_{\text{SC}}^{\text{td}}/D_v$ shows a non-linear evolution from 0.7 to 1, with values close to 1 for porosity above 0.8. The non-linearity and the high values of $D_{\text{SC}}^{\text{td}}/D_v$ comes from the contribution of the heat conduction, through $k_{\text{SC}}^{\text{C}}$, and of the latent heat, through $k_{\text{dif}}$. Finally, those estimates are in good agreement with the computed

values on 3D images of snow from Fourteau et al. (2021a).

Next, we look at the impact of temperature on the properties. The impact is weaker than the one of porosity and more complex to understand as dependencies are multiple. To help understanding, we first break down the dependencies and show in Fig. 7 how the variables $k_{\text{dif}}$ and $k_a + k_{\text{dif}}$ and the thermal conductivity of pure ice $k_i$ and pure air $k_a$ evolve with temperature. When temperature increases from 210 to 273 K, the thermal conductivity of ice decreases and the one of air slightly increase,

both evolution being quasi linear. Non-linearity is introduced with the parameter $k_{\text{dif}}$, which increases exponentially with temperature. Values for this parameter are small, even smaller than the air thermal conductivity, and are close to 0 W m$^{-1}$ K$^{-1}$ at -60°C and reach 0.02 W m$^{-1}$ K$^{-1}$ at -3°C. Finally, the term $k_a + k_{\text{dif}}$ evolves in the same way as $k_{\text{dif}}$ (non linear) but the values are increased by $k_a$.

Keeping in mind the above considerations, the evolution of $k_{\text{SC}}^{\text{eff}}$, $k_{\text{SC}}^{\text{B}}$, and $k_{\text{SC}}^{\text{C}}$ with temperature is presented in Fig. 5.b,

5.d and 5.f. For $k_{\text{SC}}^{\text{eff}}$, the SC estimates follow basically a monotonous decrease of the thermal conductivity with increasing temperature. This decrease is less pronounced for high porosity, and inversely. These features directly result from the impact of the evolution of the ice and air thermal conductivity with temperature. The evolution of $k_{\text{SC}}^{\text{B}}$ and $k_{\text{SC}}^{\text{C}}$ with temperature is more complex as the impact of $k_{\text{dif}}$ superimposes. They show non linear evolution with temperature with an evolution similar to $k_{\text{SC}}^{\text{eff}}$ for the lower temperatures transitioning to an exponential increase for the higher temperatures, the latter being driven by $k_{\text{dif}}$.

We see that this non-linearity is even more important for $k_{\text{SC}}^{\text{C}}$ than for $k_{\text{SC}}^{\text{B}}$, as $k_{\text{dif}}$ appears several time in the definition of $k_{\text{SC}}^{\text{C}}$. Finally, estimates of diffusion coefficient $D_{\text{SC}}^{\text{td}}$ show a slight influence of temperature through $k_i$, $k_a$, and $k_{\text{dif}}$ and increases with decreasing temperature, in agreement with Fourteau et al. (2021a).





### 3.3 Numerical evaluation on a simplified 2D geometry

We perform a numerical evaluation of the obtained macroscopic models on a simplified 2D snow microstructure, as in Calonne
et al. (2014b). We compare simulations of heat and water vapor transfer in snow obtained with the pore scale description and
with the macroscopic modelings.

#### 3.3.1 Case study definition

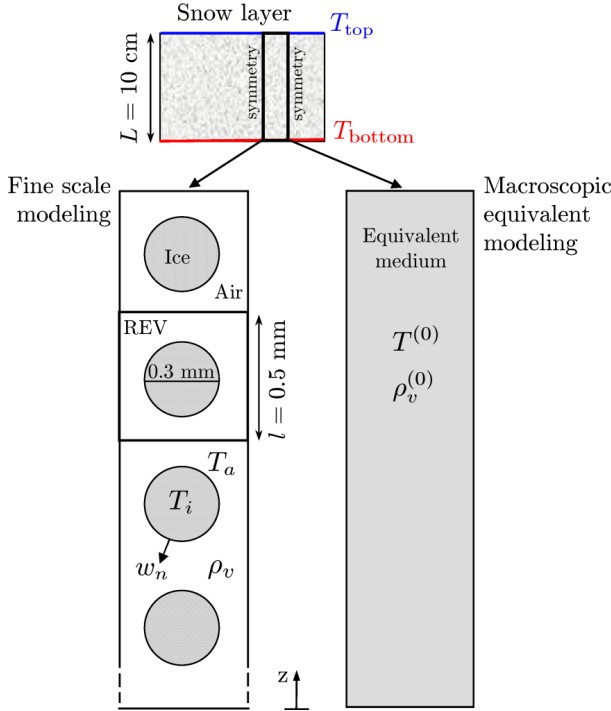

**Figure 8.** Illustration of the 2D geometry for the pore-scale modeling and the macroscopic equivalent modeling.

Finite element numerical simulations were performed using the code COMSOL Multiphysics on a 2D vertical snow layer
of 10 cm height and 0.5 cm width (Fig. 8). A constant temperature gradient of 100 K m$^{-1}$ or 500 K m$^{-1}$ are applied across
the layer. Temperature at the top $T_{\text{top}}$ and at the bottom $T_{\text{bottom}}$ are imposed and $T_{\text{bottom}}$ is kept at 273 K. For the water vapor
conditions at the top and bottom, the Robin boundary condition is applied for the pore-scale simulations and a null vapor flux is
applied for the macro-scale simulations. Symmetry conditions are imposed on the lateral sides of the snow layer. Simulations
were run in steady state.

At the pore scale, the snow layer consists in 200 periodic cells of 0.5 × 0.5 mm$^2$; each periodic cell (REV) is composed
of an ice grain of diameter 0.3 mm surrounded by air, as shown in Fig. 8. The snow porosity is 0.71, which corresponds to a
density of 266 kg m$^{-3}$. The heat and the mass transfer is described by the set of Eq. (1)-(12), where $T_i$, $T_a$, and $\rho_v$ are the





unknowns. This set of equations were numerically solved using the material parameter values presented in Table 1 and for different $\alpha$-values in the range of $10^{-10}$ to 1. For the sake of simplicity, the thermal conductivities $k_i$ and $k_a$ are at taken for -10°C and supposed to be constant in all the simulations.

At the macroscopic scale, the snow layer is seen as a continuous equivalent medium. The heat and the mass transfer is described by the homogenized equations Eq. (21) - (23) for the model A, Eq. (41) and (37) for the model B, and Eq. (54) and (45) for the model C, where $T^{(0)}$ and $\rho_v^{(0)}$ are the macroscopic unknowns. These macroscopic descriptions involve different parameters and effective properties defined over the REV, which need to be provided. The porosity and the specific surface area $\mathrm{SSA}_V$ equal to 0.71 and 3770 m$^{-1}$, respectively. The effective properties $k^{\mathrm{eff}}$ and $D^{\mathrm{eff}}$ were computed over the REV composed

of a unique cell by solving the boundary value problems (33) - (35) and (28) - (32), respectively. Given the symmetry of the REV, all the tensors involved in the macroscopic descriptions are isotropic. We found that $k^{\mathrm{eff}} = 0.04243$ W m$^{-1}$ K$^{-1}$ and $D^{\mathrm{eff}} = 1.156 \times 10^{-5}$ m$^2$ s$^{-1}$. The apparent thermal conductivity $k^B$ was analytically deduced using Eq. (41). Its value depends on temperature through the term $k_{\mathrm{dif}}(T)$. Finally, $k^C$ and $D^{\mathrm{td}}$ were computed over the REV by solving the boundary value problem (49) - (53) at different temperatures, by varying the term $k_a + k_{\mathrm{dif}}(T)$. In the considered temperature range, $D^{\mathrm{td}}$ is

almost constant and equal to $1.85 \times 10^{-5}$ m$^2$ s$^{-1}$. Figure 9 presents the evolution of $k^{\mathrm{eff}}$, $k^B$ and $k^C$ with temperature. As expected, $k^B$ and $k^C$ evolve non-linearly with $T$. To perform the simulations, the computed values of $k^C$ were fitted by the following relation: $k^C = 46.064(T/273)^4 - 156.05(T/273)^3 + 198.7(T/273)^2 - 112.68(T/273) + 24.045$.

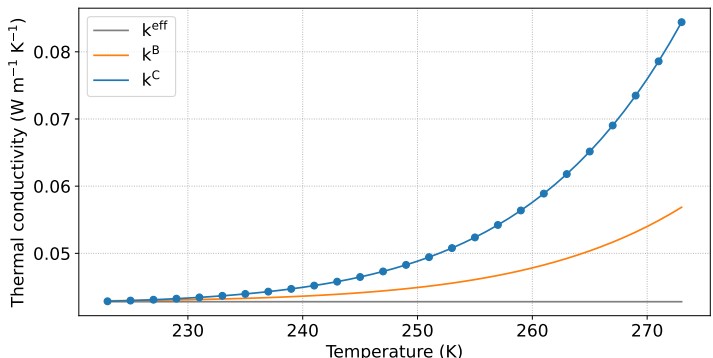

**Figure 9.** Evolution of the thermal conductivities $k^{\mathrm{eff}}$, $k^B$ and $k^C$ with temperature. For $k^C$, the blue dots represent the numerical estimates of $k^C$ and the blue line is the fit.

### 3.3.2    Comparison between pore-scale and macro-scale simulations

Results between pore-scale and macro-scale simulations are compared in terms of temperature, vapor density, and mass change

rate. At the pore-scale, the average values of each variable were taken over the cell and computed as follows:

$$\langle T \rangle = \frac{1}{\Omega}\left(\int_{\Omega_i} T_i d\Omega + \int_{\Omega_a} T_a d\Omega\right), \quad \langle \rho_v \rangle = \frac{1}{\Omega_a}\int_{\Omega_a} \rho_v d\Omega \quad \langle \rho_{vs}(T) \rangle = \frac{1}{\Omega_a}\int_{\Omega_a} \rho_{vs}(T_a) d\Omega \quad \langle \dot{\phi} \rangle = \frac{1}{\Omega}\int_{\Gamma} w_n d\Gamma \tag{72}$$

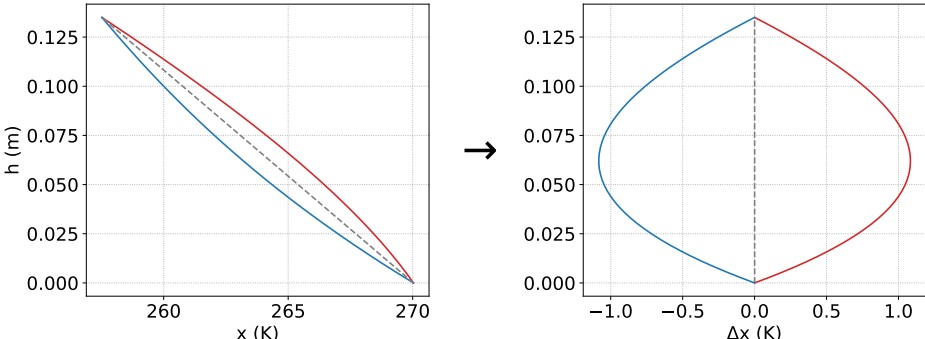

**Figure 10.** Simplified example of the transition from the x to $\Delta$x notation in a concave (red), and a convex (blue) case.

Thus, we compare the vertical profiles of the pore-scale variables $\langle T \rangle$, $\langle \rho_v \rangle$, $\langle \rho_{vs} \rangle$ and $\langle \dot{\phi} \rangle$ with the vertical profiles of the macroscopic variables $T^{(0)}$, $\rho_v^{(0)}$, $\rho_{vs}^{(0)}(T^{(0)})$ and $\dot{\phi}$. As the obtained simulated temperature profiles were close to each other, to ease the comparison, we also use the temperature deviation $\Delta T$, which represents the deviation of the simulated temperature

profile from the linear temperature profile imposed by $T_{\text{top}}$ and $T_{\text{bottom}}$, as illustrated in Fig. 10. In the same vein, we use the water vapor supersaturation, which is the difference between the simulated water vapor density and the saturation water vapor density $\rho_v - \rho_{vs}(T)$. Figure 11 shows the vertical profiles of $\Delta T$, of $\rho_v - \rho_{vs}(T)$, and of $\dot{\phi}$ from the pore-scale simulations (dots) and the macroscopic models (lines), considering a temperature gradient of 100 and 500 K m$^{-1}$. For the pore-scale simulations, values of $\alpha$ from $10^{-9}$ to 1 were used. For the macroscopic models, results are only shown in their domain of

validity with respect to $\alpha$. To further highlight the impact of $\alpha$, Fig. 12 presents the evolution of $T$, $\rho$ and $\rho_{vs}$ with $\alpha$ for a specific cell of the snow layer, here the hundredth cell from the bottom ($x = l/2, y = 100l - l/2$). Again, pore-scale simulations (dots) and the macroscopic models (line) are compared.

We describe first the main features observed in the pore-scale simulations. All the variables show an impact of the $\alpha$-value. The temperature deviation $\Delta T$ is overall mainly positive (Fig. 11.a and b), which reflects the presence of a heat source

by non-conductive processes such as latent heat from deposition. This temperature deviation increases with $\alpha$ and with the temperature gradient. This is also reflected in the temperature of the middle cell that overall increases with increasing $\alpha$ (Fig. 12.a and b). This increase is not uniform and two plateau are observed, one between $10^{-6} \leqslant \alpha \leqslant \alpha_T$, and the other one between $10^{-1} \leqslant \alpha \leqslant 1$. The largest $\Delta T$ value is reached in the center of the snow layer and is around 0.4 K at 100 K m$^{-1}$ and 4 K at 500 K m$^{-1}$. Looking at the lower part of the layer, negative $\Delta T$ values can be found for $\alpha \leqslant \alpha_T \sim 3 \times 10^{-4}$ and indicate

a heat loss by non-conductive processes such as latent heat from sublimation. This feature vanishes for the large temperature gradient. In terms of water vapor supersaturation $\rho_v - \rho_{vs}(T)$, we observe positive values (over-saturation) in the upper part of the snow layer, values close to zero in the central part (at saturation), and negative values (under-saturation) in the lower part (Fig. 11 c, d). The largest over-saturation and under-saturation values are shown for low $\alpha$-values, when phase changes are very limited. With increasing $\alpha$-values, values close to saturation gets predominant and the over-saturation and under-saturation

zones become localized near the top and bottom, respectively. This is confirmed in Fig. 12.c and 12.d. where $\rho_v \approx \rho_{vs}(T)$ for



**Figure 11.** Vertical profiles of $\Delta T$, $\rho - \rho_{vs}(T)$, and $\dot{\phi}$ from the pore scale simulations (dots) and from the macroscopic model A (grey lines), B (orange lines), and C (blue lines), considering a temperature gradient of 100 and 500 K m$^{-1}$ and for different values of $\alpha$. $\Delta T$ represents the deviation of the temperature profile from a linear temperature profile.





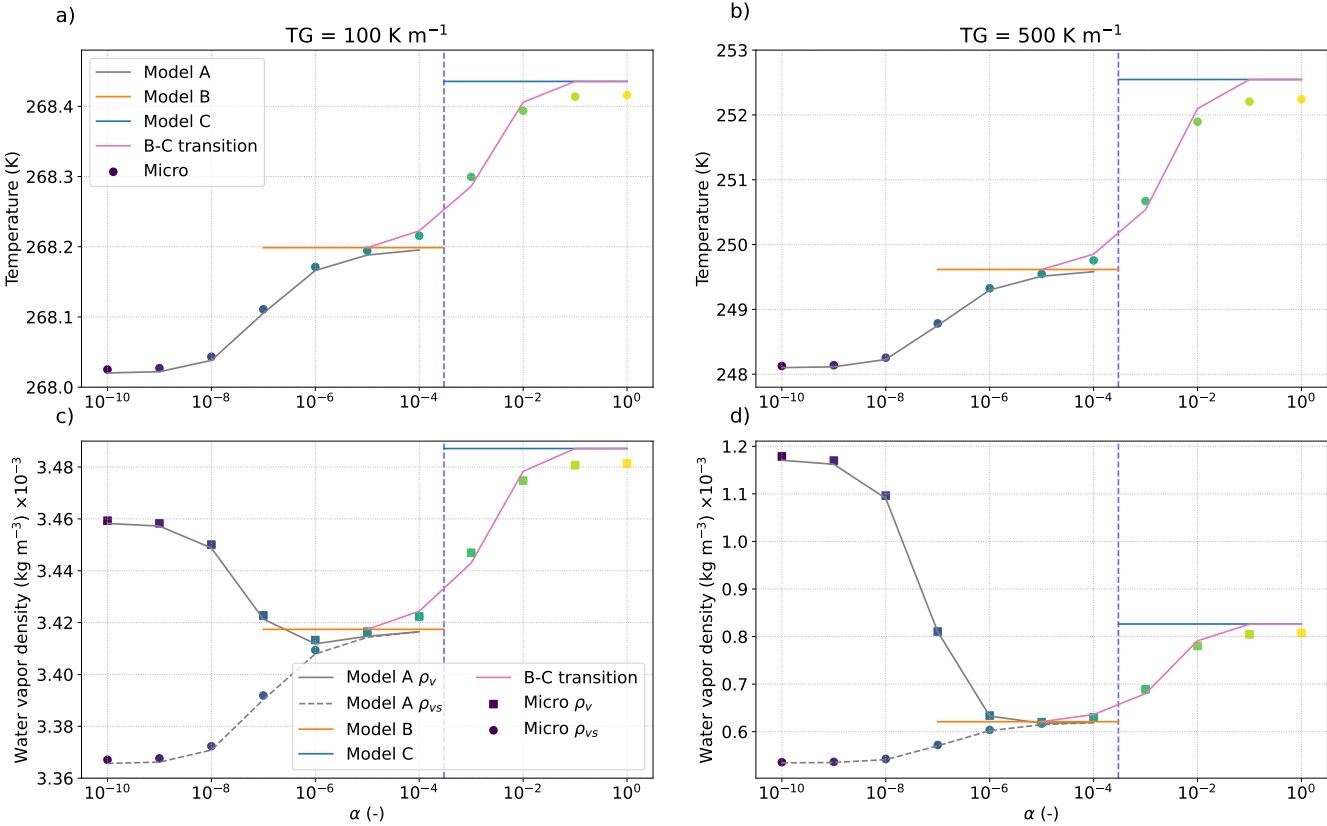

**Figure 12.** Temperature and water vapor density in the middle of the snow layer as a function of $\alpha$, obtained from the pore-scale simulations (dots) and from the macroscopic model A (grey lines), B (orange lines), and C (blue lines), at 100 and 500 K m$^{-1}$. The models are only shown for the $\alpha$-values within their domain of validity. Values of saturation water vapor density $\rho_{vs}$ from the pore-scale simulations and from the model A are also presented. The pink curve represents the transition between the model B and C presented in Sect. 5.

$10^{-6} \leqslant \alpha \leqslant 1$. Similarly to temperature, two plateau are shown where $\rho_v - \rho_{vs}(T)$ evolve little with $\alpha$. All these results are consistent with our theoretical analysis presented in Sect. 2. Last, the vertical profiles of $\dot{\phi}$ are consistent with the ones of supersaturation, showing deposition in the upper part where the porosity decreases and sublimation in the lower part where the porosity increases (Fig. 11.e and f). As $\alpha$ increases, those transitions become sharper and sharper, like a front. For $\alpha_T \leqslant \alpha \leqslant 1$,

most values become negative, indicating overall deposition in the snow layer. A sublimation zone is still visible at the bottom of the snow layer but its thickness is typically of the order of a few REV or smaller. Finally, as the difference $\rho_v - \rho_{vs}$ is directly related to the interface growth velocity $w_n$ (see Eq. 10), and as it could be useful to compare it with experimental estimates (e.g., Flin and Brzoska, 2008; Brzoska et al., 2008; Pinzer et al., 2012; Libbrecht and Rickerby, 2013), we provide below the mean values of $w_n$ computed over the bottom and middle cell for $\alpha = 10^{-6}$. For 100 K m$^{-1}$, a value of 5.9 $\times 10^{-13}$ m s$^{-1}$ and





**Table 2.** Overview of the experimental settings used in the simulations.

| Experiment | Bouvet A | Bouvet B | Kamata |
|---|---|---|---|
| initial density (kg m$^{-3}$) | 210 | 287 | 165 |
| snow layer height (cm) | 13.5 | 7.7 | 10 |
| temperature gradient (K m$^{-1}$) | 93 | 103 | 530 |
| snow base temperature (°C) | -3.1 | -6.5 | -12 |
| snow surface temperature (°C) | -15.6 | -14.5 | -65 |
| duration (days) | 20 | 28 | 5.5 |

of -2.7 $\times 10^{-11}$ m s$^{-1}$ is found in the middle and bottom cell, respectively. For 500 K m$^{-1}$, a value of 4.5 $\times 10^{-12}$ m s$^{-1}$ and of -1.1 $\times 10^{-10}$ m s$^{-1}$ is found in the middle and bottom cell, respectively.

Next we compare the different macroscopic models to the pore-scale simulations. In both Fig. 11 and Fig. 12, the comparison shows different behaviors depending on $\alpha$. For $\alpha \leqslant \alpha_T$, the model A reproduces precisely all the features shown at the pore-scale. The model B and C are independent of $\alpha$ and provide one estimate of the temperature, and thus the water vapor density,

for all the $\alpha$-values in their domain of validity. These estimates are only able to reproduce the plateau values observed in the pore-scale simulations, i.e. temperatures for $10^{-6} \leqslant \alpha \leqslant \alpha_T$ for the model B and $10^{-1} \leqslant \alpha \leqslant 1$ for the model C (Fig. 12). Both models B and C predict only deposition in snow, with negative $\dot{\phi}$ values throughout the layer (Fig. 11.e and f). They do not capture the sublimation front at the bottom of the snow layer, in contrast with the model A. The mass balance between sublimation-deposition over the whole snow layer is not well satisfied: the Dirichlet boundary condition on the temperature

field at the bottom and the top of the snow cannot ensure that the water vapor flux is null at the same time since $\rho_v = \rho_{vs}(T)$. In order to overcome this limit, specific boundary conditions should be introduced to allow describing mass variations near the interfaces.

## 4 Application to experimental data

This section presents the evaluation of the macroscopic models A, B and C based on observations of natural snow evolution

from three cold-laboratory TGM experiments. We first introduce the experimental data (Sect. 4.1), then we define the estimates to be taken for the input parameters of the models (Sect. 4.2), and, finally, we present the simulation results with the models and their comparison with the experiments (Sect. 4.3).

### 4.1 Experimental datasets

We used the datasets provided by Bouvet et al. (2023), consisting of two experiments referred as Bouvet A and Bouvet B in

their paper and hereafter, and the data from Kamata and Sato (2007), referred as 'Kamata'. These experiments provide the





required data to evaluate our models: time-series of the vertical profiles of temperature and density of a snow layer evolving under a temperature gradient in a controlled environment. The main characteristics of the three experiments are summarized in Table 2. Bouvet A is a TGM experiment on a 13.5 cm height snow layer for which a TG of 93 K m$^{-1}$ was applied during 20 days. X-ray tomography was done at regular time intervals resulting in 9 large 3D images of the whole vertical dimension

of the snow layer at a resolution of 21 $\mu$m and 17 small 3D images of the top or bottom part of the layer at a resolution of 8 $\mu$m. For the large images, the first few mm at the base of the layer is lacking, due to the snow sampling procedure, so no data are available for this area. This experiment also includes monitoring of the temperature profile of the snow layer, measured using 7 PT100 sensors. Bouvet B is a TGM experiment on a 7.7 cm height snow layer for which a TG of 103 K m$^{-1}$ was applied during 28 days. Four tomography images of the first lower 4.2 cm of the snow layer are provided at a resolution of

10 $\mu$m. For both experiments, Bouvet A and Bouvet B, the vertical profiles of snow density computed from the 3D images are provided and vertical mass redistribution can be analyzed. Finally, the Kamata experiment is a TGM experiment on a snow layer of 10 cm height for which an extreme TG of 530 K m$^{-1}$ was applied during 5.5 day (133 hours). The vertical mass redistribution was estimated by measuring snow density for four sections of the snow layer. For that, the snow layer was separated in 4 compartments of about 2.5 cm height each using horizontal nylon meshes, which enables water vapor to

get through. Each compartment was weighed at the initial and final stage of the experiment. In addition, the temperature was recorded at 6 vertical locations.

## 4.2 Effective properties and parameters

Next we study the estimates of the effective properties and others input parameters required to run the model A, B and C. For each model, these properties are computed from the 3D images of snow of the experiment Bouvet A and Bouvet B. Those

values are then compared to different parameterizations from the literature or fitted regressions and we select the more suited ones to be used later in the models.

### 4.2.1 Model A

The model A involves three effective parameters that are the effective thermal conductivity $k^{\text{eff}}$, the effective vapor diffusivity $D^{\text{eff}}$ and the SSA (Sect. 2.6.1). These parameters were estimated in the case of Bouvet A and Bouvet B by numerical compu-

tations on the 3D tomographic images available. SSA was computed per unit of mass based on the voxel projection approach (Flin et al., 2011; Dumont et al., 2021). $k^{\text{eff}}$ and $D^{\text{eff}}$ were computed with the software Geodict (Thoemen et al., 2008) by solving the boundary value problems (28) - (32) and (33) - (35) on the 3D images, applying periodic boundary conditions on the external boundaries, as described in Calonne et al. (2011, 2014b). Values of $k_{\text{i}}$ and $k_{\text{a}}$ at -10°C for the computation of $k^{\text{eff}}$. The obtained 3D tensors of both properties show negligible non-diagonal terms. In the following, we refer to $k^{\text{eff}}$ and $D^{\text{eff}}$ as

the average of the diagonal terms of the tensors.

Figure 13 presents the results of the image-based computations of $k^{\text{eff}}$, $D^{\text{eff}}$ and SSA, for the experiment of Bouvet A and Bouvet B. To compare with, we show the estimates of $k^{\text{eff}}$ and $D^{\text{eff}}$ by the SC model presented in Section 3.2, the density-based parameterizations of $k^{\text{eff}}$ from Calonne et al. (2011) and Riche and Schneebeli (2013), and a fitted regression of the SSA





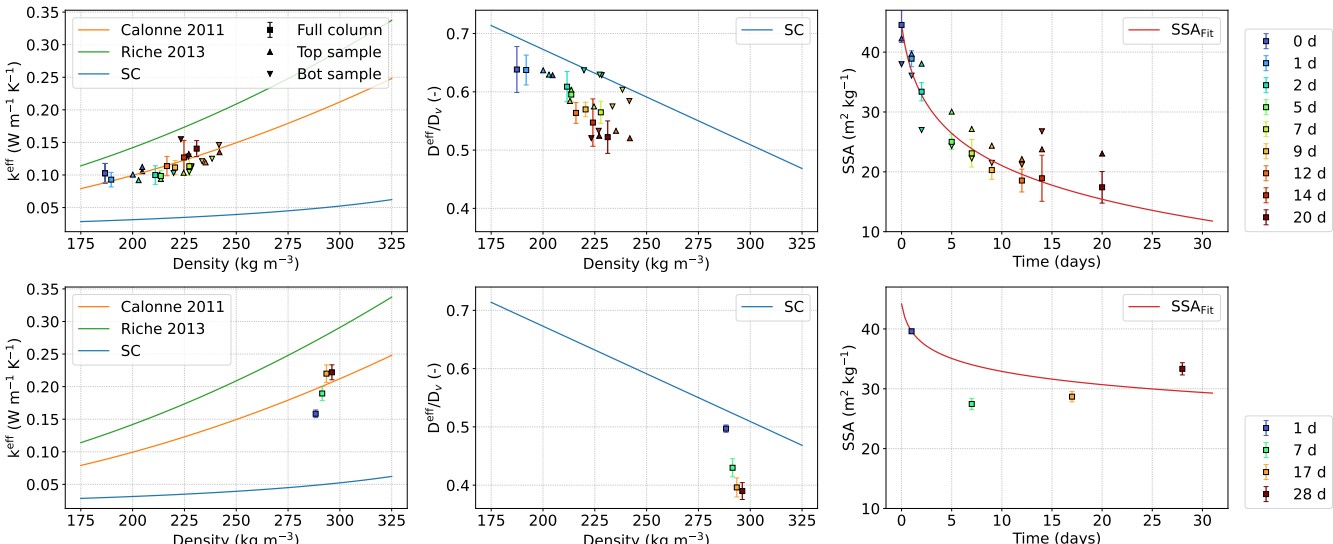

**Figure 13.** Average values of effective conductivity and normalized effective diffusivity as a function of density, and SSA as a function of time, computed from the tomography images of Bouvet A (symbols, upper plots) and Bouvet B (symbols, lower plots). The error bars represent the standard deviation of the parameter along the image height. Comparison with the SC model, classical parameterizations, and fits are shown (solid lines).

values as a function of time, referred as $SSA_{Fit}(t)$, based on a logarithmic function as formerly proposed by Legagneux et al.

(2004). For thermal conductivity, the parameterization of Calonne et al. (2011) are in good agreement with the image-based computations in both experiments, whereas the parameterization of Riche and Schneebeli (2013), which specifically describes the case of depth hoar, predicts slightly larger values. The SC model largely underestimates the values, about two to four times smaller than the image-based computations. For the vapor diffusion coefficient, the SC model provides overall fair estimates, which are slightly overestimated, especially towards the end of the experiments, as reported for depth hoar and faceted crystals

in Calonne et al. (2014b). For SSA, the fit reproduces well the SSA evolution for the experiment Bouvet A. For Bouvet B, SSA does not follow the classic exponential decrease but it increases after 7 days and until the end of the experiment; this increase is specific to hard depth hoar formation (Bouvet et al., 2023). This feature is not predicted by the applied fit, yet it provides fair estimates of the SSA values.

Given the above considerations, we selected two sets of parameters to simulate Bouvet A, Bouvet B, and Kamata with the

model A, which are summarized in Table 3. In the set 'Calonne', $k^{eff}$ is estimated with the parameterization of Calonne et al. (2011). In the second set 'SC', $k^{eff}$ is given by the self-consistent estimates. In both sets, $D^{eff}$ is estimated with the SC model and the SSA with the logarithmic fit, which is specific for Bouvet A and Bouvet B. In the case of the Kamata experiment, we cannot test the proposed estimates of $k^{eff}$, $D^{eff}$ and SSA against reference data, as such data are not available from Kamata and Sato (2007). SSA evolution was reproduced based on the logarithmic fit from Bouvet A, as both experiments are the closest

in terms of initial snow type, grain size and density. In the model evaluation that follows (Sec. 4.3), the set 'Calonne' is the





**Table 3.** Summary of the effective parameters used in the simulations.

| | | |
|---|---|---|
| **Model A** | Set SC | $k_{SC}^{eff}, D_{SC}^{eff}, SSA_{Fit}(t)$ |
| | Set Calonne | $k_{Calonne}^{eff}, D_{SC}^{eff}, SSA_{Fit}(t)$ |
| **Model B** | Set SC | $k_{SC}^B = k_{SC}^{eff} + k_{dif}D_{SC}^{eff}/D_v, D_{SC}^{eff}$ |
| | Set Calonne | $k_{Calonne}^B = k_{Calonne}^{eff} + k_{dif}D_{SC}^{eff}/D_v, D_{SC}^{eff}$ |
| **Model C** | Set SC | $k_{SC}^C, D_{SC}^{td}$ |
| | Set Fit | $k_{Fit}^C, D_{SC}^{td}$ |

one per default used to evaluate the model A. Results with the set 'SC' are also presented to illustrate an alternative choice of parameters, which, although less accurate, allows for consistent and analytically-based estimates for all properties.

### 4.2.2 Models B and C

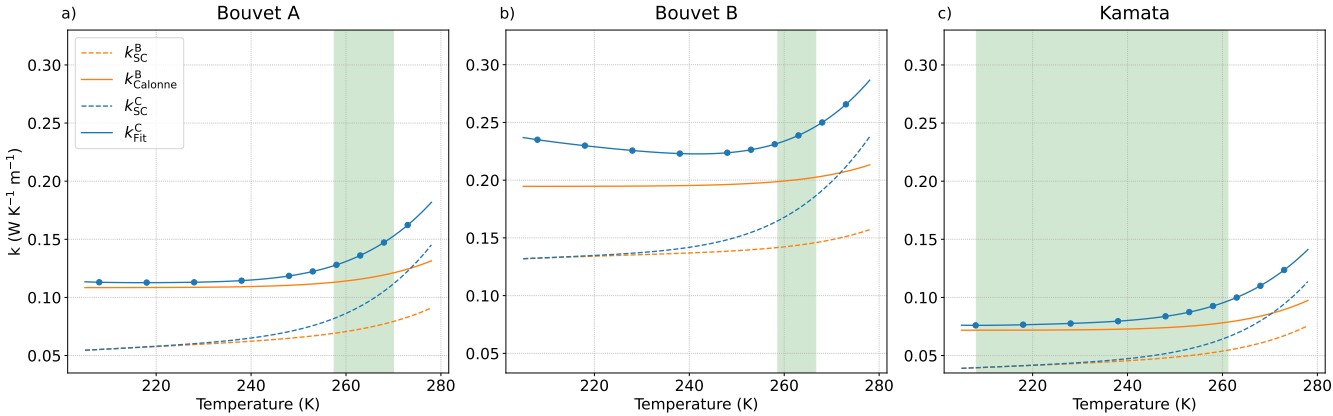

**Figure 14.** The thermal conductivity estimates for the model B $k_{SC}^B$ and $k_{Calonne}^B$, and for the model C $k_{SC}^C$ and $k_{Fit}^C$, are presented as a function of temperatures (lines). The parameters are presented for Bouvet A, Bouvet B and Kamata experiment. The computed values on 3D images used to derived $k_{Fit}^C$ are shown by blue dots. The green areas represent the temperature ranges of each experiment.

The models B and C only involve the apparent thermal conductivities of snow $k^B$ and $k^C$, respectively. Estimating $k^B$ comes

down to estimating $k^{eff}$ and $D^{eff}$, as it is defined as $\mathbf{k}^B = \mathbf{k}^{eff} + k_{dif}\mathbf{D}^{eff}/D_v$. For that, we use the same estimates of $k^{eff}$ and $D^{eff}$ selected for the model A as described above. So two sets of input parameters were used for the model B: the set 'Calonne', from which the model's performances are evaluated, and the alternative set 'SC' (Tab. 3). Figure 14 presents the evolution of $k_{Calonne}^B$ and $k_{SC}^B$ with temperature for each experiment, taking the mean snow density of the experiments (see Tab. 2). Both





estimates show similar trend but, as in the model A, the SC estimate predicts lower values than when using the parameterization

of Calonne et al. (2011).

For the model C, $k^C$ was computed on the 3D images from Bouvet A and Bouvet B, by solving the boundary problem (49) - (53) using the Geodict software. As above, only diagonal terms of the tensor were considered and $k^C$ refers to the average value of the diagonal terms. Here, computations were performed on only one REV from each experiment and for 10 temperatures ranging from 210 to 273 K. We selected the image at 14 days for Bouvet A (cropped between 5.8 and 6.7 cm height) and the

image at 7 days for Bouvet B (cropped between 1.7 and 2.5 cm height). To be able to estimate $k^C$ for the Kamata experiment, we took a 3D image of snow with similar characteristics and used the one from Fourteau et al. (2021a) of depth hoar with a density of 165 kg m$^{-1}$. Results of the image-based computations are presented in Fig 14, as well as estimates from the SC model $k^C_{\mathrm{SC}}$ presented in Sec. 3.2. A fit on the computed data is also shown and refers as $k^C_{\mathrm{Fit}}$. Again, the SC estimates for the model C captures the trend but largely underestimate the values. In what follows, simulations were performed using the

fitted values $k^C_{\mathrm{Fit}}$, referred as the set 'Fit' in Tab. 3, from which the evaluation of the model C is based on. As for the other models, simulations with the SC estimates $k^C_{\mathrm{SC}}$ are also presented for a sake of comparison and as they allow for independent and consistent estimates.

## 4.3 Comparisons between models and experiments

In this section we compare simulations from the models A, B and C with the measurements from the three experiments

Bouvet A, Bouvet B, and Kamata. The simulations were performed with the software COMSOL Multiphysics by resolving the homogenized equations on a 1D geometry that corresponds to the snow layer of experiments. Equations are Eq. (21) - (22) for the model A, Eq. (41) and (37) for the model B, and Eq. (54) and (45) for the model C. For the model A, the boundary conditions in temperature are the top and bottom imposed temperatures of the experiments. In terms of vapor density, the conditions correspond to zero flux at the top and bottom. Additionally, the source term is forced to zero in the simulation

nodes where a density of zero is reached. For the models B and C, the boundary conditions are the imposed temperatures. The models were run using the sets of input parameters described in Table 3 and and considering the experimental conditions summarized in Table 2. Comparisons between measurements and simulations are performed based on temperature and mass change variables.

### 4.3.1 Temperature

Figure 15 presents the measured and simulated vertical profiles of $\Delta T$ with the models A, B and C for the three experiments, taking different $\alpha$-values from $10^{-9}$ to $10^{-4}$ for the model A. The $\Delta T$ values analyzed here are the ones computed at the beginning of the experiments when the temperature gradient is well established but before the formation of the air gap. Overall, profiles of $\Delta T$ are of similar shapes as the ones simulated on the simplified 2D microstructure (Sec. 3.3), describing right-headed curves indicating that processes apart from pure heat conduction, such as phase change, occur and result in a heat source

in the snow layer. In the center part of the layer, a maximum deviation of 1.15 K was measured in the Bouvet A experiment



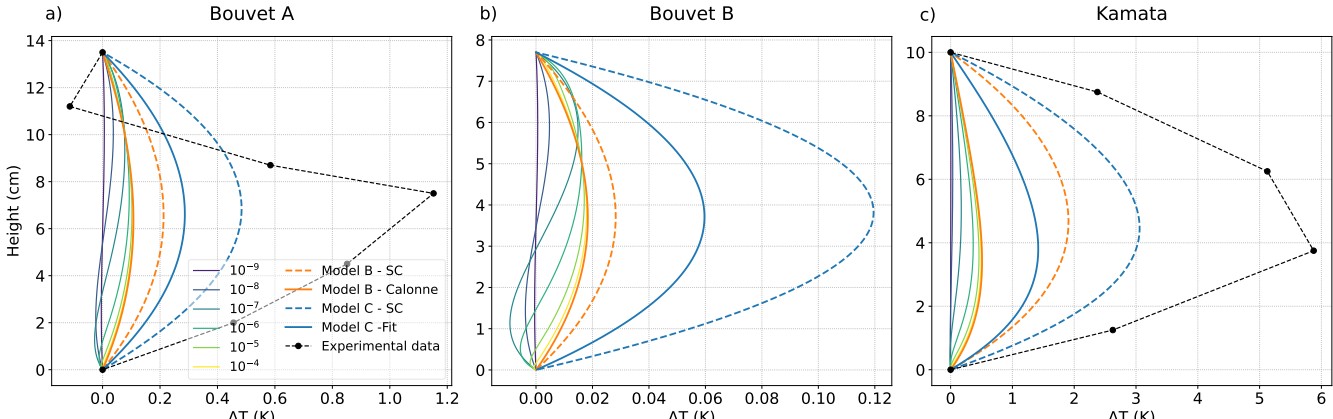

**Figure 15.** Vertical steady state profiles of $\Delta T$ simulated with the model A with $\alpha$ ranging from $10^{-9}$ to $10^{-4}$, with the model B and the model C, for Bouvet A, Bouvet B and Kamata experiments. Simulations using the set 'Calonne' (solid lines) are shown for the model A and using both set 'Calonne' (solid lines) and set 'SC' (dashed lines) for the models B and C. The experimental profiles are shown with black dashed lines for Bouvet A and Kamata.

and of 5.9 K in the Kamata experiment. The negative $\Delta T$ value in the upper part of the layer in Bouvet A is attributed to a temperature sensor error (Bouvet et al., 2023).

Looking at the models, the main observation is that they all underestimate $\Delta T$. In more details, the model A predicts negative $\Delta T$ in the lower part of the snowpack and positive otherwise, reflecting a heat sink attributed to more sublimation in the lower 645 part and a heat source attributed to more deposition in the rest of the layer. With increasing $\alpha$, the positive values of $\Delta T$ increase and the negative ones tends to vanish, so that the shape of the simulated curve become closer to the experimental one. The maximum $\Delta T$ predicted by the model A is reached for the highest $\alpha = 10^{-4}$ and is of 0.11 K for Bouvet A and of 0.49 K for Kamata, which corresponds only to 10% and 8% of the experimental value, respectively.

The models B and C show a unique $\Delta T$ profile valid over their domain of validity, $10^{-6} \leqslant \alpha \leqslant \alpha_T$ and $\alpha_T \leqslant \alpha \leqslant 1$, 650 respectively. The profile shape is in agreement with the measurements, showing only positive values throughout the layer. $\Delta T$ values of the model B correspond to the upper limit of the model A. The model C is the closest to the experimental data. Still, values are largely underestimated and reach at most 0.29 K (25% of the experimental data) for Bouvet A and 1.4 K (23% of the experimental data) for Kamata. In both models B and C, slightly better results are found when using the 'SC' set of input parameters, even though it corresponds to underestimated estimates as seen in Sec. 4.2. This better agreement with the SC 655 estimate is somehow artificial and comes from the fact that the lower values of $k_{\mathrm{SC}}^{\mathrm{B}}$ and $k_{\mathrm{SC}}^{\mathrm{C}}$ compared to $k_{\mathrm{Calonne}}^{\mathrm{B}}$ and $k_{\mathrm{Fit}}^{\mathrm{C}}$, respectively, lead to reduce the overall heat conduction through snow and allow for higher $\Delta T$, as well as the fact that the SC estimates allow for a slightly higher sensitivity (steeper slope) of the thermal conductivity to temperature in the temperature range of the considered experiments (see the green areas in Fig. 14). A final interesting point is the strong impact of the density on $\Delta T$, which can be seen by comparing simulations of Bouvet A and Bouvet B, for which temperature gradients were very 660 close but snow density was 210 and 287 kg m$^{-3}$, respectively. For the same temperature gradient, the higher the snow density,





the higher the heat conduction through snow and the lower the $\Delta T$. For example, in lighter snow (Bouvet A), a maximum $\Delta T$ of 0.29 K is predicted by the model C against 0.06 K for the denser snow (Bouvet B).

### 4.3.2 Mass change

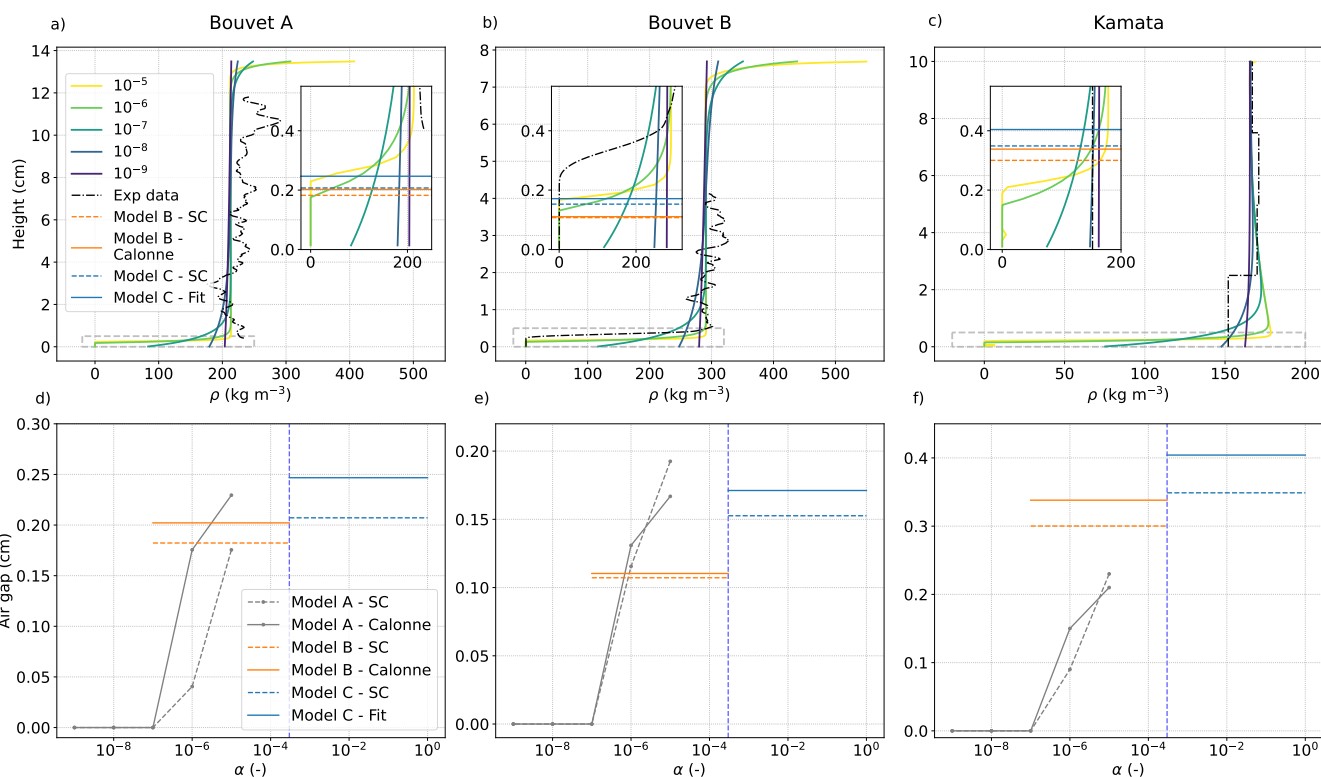

**Figure 16.** (a ,b, c): Density profiles from the macroscopic models and from the experimental data for the final stage of the Bouvet A, Bouvet B and Kamata experiments. Results of the model A are provided for $\alpha$-values from $10^{-5}$ to $10^{-9}$ and for the parameter set 'Calonne'. The height of the air gaps derived for the models B and C are shown with horizontal bars in the zoom boxes. Results of the model B are provided for the set 'SC' (orange dashed lines) and the set 'Calonne' (orange solid lines). Results of the model C are provided for the set 'SC' (blue dashed lines) and the set 'Fit' (blue solid lines). (d, e, f): Air gap height at the final stage of the experiments as a function of $\alpha$, simulated with the models A, B and C using the parameter sets 'SC', 'Calonne', and 'Fit'.

Next we evaluate the models regarding mass changes across the vertical dimension of the snow layer. We look at the vertical 665 density profile of snow, as well as the height of the air gap formed at the base of the layer at the end of the experiments, caused by an upward mass transfer during TGM. For the model A, the air gap height is defined as the highest height value at which the density is zero. For the models B and C, the vertical profile of density cannot be evaluated because they only predict deposition and thus density increase, due to boundary condition issue as already described in Sect. 3.3. Still, to allow for a comparison with measurements, we derived a rough estimate of the air gap by considering that all the mass gain in the snow layer over the





whole experiment duration is balanced by a mass loss localized at the very bottom of the snow layer, leading to a sharp air gap described as:

$$h_{\text{air gap}} = H \frac{\overline{\dot{\phi}} \, t_{\text{exp}}}{(\overline{\dot{\phi}} \, t_{\text{exp}} + \phi_{\text{init}} - 1)} \quad \text{with} \quad \overline{\dot{\phi}} = \frac{1}{H} \int_0^H \dot{\phi}(z) dz = -\frac{1}{H} \int_0^H \frac{1}{\rho_i} \frac{\partial}{\partial z} \left( \frac{\partial D \rho_{vs}^{(0)}(T)}{\partial z} \right) dz \quad (73)$$

with $h_{\text{air gap}}$ the height of the air gap (m), $H$ the total height of the snow layer (m), $\phi_{\text{init}}$ the initial porosity (-) and $t_{\text{exp}}$ the total duration of the experiment (s) (Table 2). $D$ is the diffusivity coefficient and corresponds to $D_{\text{SC}}^{\text{eff}}$ for the model B and $D_{\text{SC}}^{\text{td}}$

for the model C.

Figure 16 shows the vertical profile of density and the height of the air gap simulated and measured in Bouvet A, Bouvet B, and Kamata at the end of the experiments. The Bouvet B and Kamata experiment report a mass loss in the lower part of the snow layer. In Bouvet B, it results in the formation of an air gap of 2.7 mm height at the layer base, at which the snow density drops from about 290 to 0 kg m$^{-3}$ within a few mm (Fig. 16.b). In Kamata, the initial uniform density profile around 165 kg

m$^{-3}$ evolved and show at the final stage a density of 152 kg m$^{-3}$ at the bottom of the layer which is lower than elsewhere, where density is around 170 kg m$^{-3}$ (Fig. 16.c). So only a decrease in density at the base was observed, not an air gap. This might be however prevented by the vertical resolution of the density measurement of 2.5 cm in Kamata, at which the detection of a mm-scale air gap is not possible. To provide an estimation of the height of the potential air gap, we converted the density decrease in the bottom first 2.5 cm into a pure air gap. This would lead to a 2.6 mm height air gap, similar to the one measured

for Bouvet B for a much lower temperature gradient. Finally, as already mentioned, the experiment of Bouvet A does not include the first mm at the base of the snow layer, so comparison with simulations is not possible.

We look at the experiments Bouvet A and Bouvet B and describe first the simulations of the model A, performed with $\alpha$ from $10^{-9}$ to $10^{-5}$ (at $\alpha = 10^{-4}$ the model becomes numerically unstable). The model predicts similar mass transport for both experiments: a mass gain in the upper part of the layer and a mass loss in the upper part, the latter feature being consistent with

the measurements. In more details, and as for temperature, the impact of $\alpha$ is clearly shown. For the lowest $\alpha$-value of $10^{-9}$, the density profile is almost linear so the mass redistribution is even throughout the layer. As $\alpha$ increases, the area of mass loss and mass gain become more localized near the base and top of the layer and the density transitions become sharper. From $\alpha = 10^{-6}$ and above, an air gap is simulated with density values reaching 0 kg m$^{-3}$ at the bottom. The air gap closest to the experiments is obtained with the highest $\alpha = 10^{-5}$ and reaches 2.3 mm height for Bouvet A and 1.65 mm for Bouvet B, which

corresponds fairly well to the measured air gap, yet slightly underestimated (75%) (Fig. 16.d and 16.e). Approximations of the air gap for the models B and C are close to the ones simulated by the model A, so that all the models seem to underestimate the air gap. For Bouvet B, an air gap of 1.7 mm is estimated for the model C (63% of the experimental air gap) and of 1.1 mm for the model B (41% of the experimental air gap). Finally, for all the models, using the alternative 'SC' set of input parameters (Tab. 3) has little impact on the air gap and leads mostly to a slight reduction in height (dashed lines in Fig. 16).

Simulations of the Kamata experiment with the model A differ from the ones of Bouvet A and Bouvet B. Indeed, a mass gain is not predicted in the upper part of the snow layer but instead in a zone right above the mass loss region. This is particularly visible for $\alpha = 10^{-5}$, where the air gap, located in the first 2.1 mm, is directly surmounted by the densest part of the snow layer, located around 4 mm, with a density reaching 175 kg m$^{-3}$. Simulations seem to show that mass redistribution in the Kamata



experiment occur mostly in the lower part of the snow layer, which might be due to the impact of temperature on the simulated
heat and mass transport processes so that they are reduced in the very cold upper part (-65°C). This effect of temperature can
also be seen in the $\dot{\phi}$ simulations on the simplified microstructure for the temperature gradient of 500 K m$^{-1}$, for which the
imposed temperature conditions were close to the ones of Kamata, as presented in Fig. 11.f. Considering that this effect applies
in reality, it would imply that the final density measured in Kamata in the bottom first 2.5 cm is the result of both the mass loss
and mass gain and thus that the above estimation of an air gap of 2.6 mm might be underestimated. Comparisons of air gaps
for the Kamata experiment should be looked at with the above consideration in mind. When averaging the simulated density
values of the model A over a 2.5 cm step, as done in the measurements, a value of 158 kg m$^{-3}$ is found for the first 2.5 cm, in
agreement with the measured one of 152 kg m$^{-3}$. Coming back to the air gap comparisons, the model A predicts well the air
gap estimated for the Kamata experiment when the highest $\alpha$-value is considered. At $\alpha = 10^{-5}$, the simulated air gap is of 2
mm, again close to the estimated one of 2.6 mm, yet underestimated (77%). Unlike for Bouvet A and Bouvet B, the models B
and C stand out from the model A and their approximations of the air gap are significantly larger, between 3 mm and 4 mm.
They would thus predict larger air gap than the one from the experiment, up to twice the height.

## 5   Discussion

### 5.1   Modeling heat and mass transfer with the models A, B or C

In the present work, macroscopic models for heat and mass transfer in dry snow have been derived by homogenization from
the physics at the pore scale for different values of the condensation coefficient $\alpha$ in the range [$10^{-10}$, 1]. The latter was
assumed to be constant in the whole modeled snow layer. The Robin boundary equation for the water vapor at the ice-air
interface allowed to define a transition value $\alpha_T$, which equals $\approx 3 \times 10^{-4}$ for typical snow grain size around 0.5 mm, that
characterizes the transition between the two main mechanisms driving the water vapor transfer through the snowpack: diffusion
and sublimation-deposition. The homogenization process allowed (i) to retrieve three different models already proposed in the
literature (Calonne et al., 2014b; Hansen and Foslien, 2015; Moyne et al., 1988) and to specify their domains of validity
according to the $\alpha$-values, and (ii) to show that the hypothesis $\rho_v = \rho_{vs}(T)$, which is often made, is a good approximation for
$\alpha$-values larger than $10^{-6}$.

   At the macroscopic scale, the model A (Calonne et al., 2014b), valid for $\alpha$-values in the range [$10^{-10}$, $\alpha_T$], is described
by two coupled equations, one for the temperature field and one for the water vapor field. They are coupled by a source term
that reflects the sublimation-deposition process and depends on $\alpha$. In this model, the induced porosity variation in the snow
layer can be easily computed. In the case of the model B and the model C (Moyne et al., 1988; Hansen and Foslien, 2015),
the physics at macro-scale is driven by the temperature field only as $\rho_v = \rho_{vs}(T)$. Because the models only solve temperature
field, it is not as straightforward to access the porosity variation. In our case, both models do not satisfy mass conservation and
predict only deposition over the whole snow layer and so the sublimation front occurring at the bottom of the snow layer, as
seen in the comparison between pore-scale and macroscopic-scale simulations (Fig. 11). In the future, a more reliable boundary
condition, as a Stefan boundary condition, should be introduced to better describe the evolution of the sublimation front.





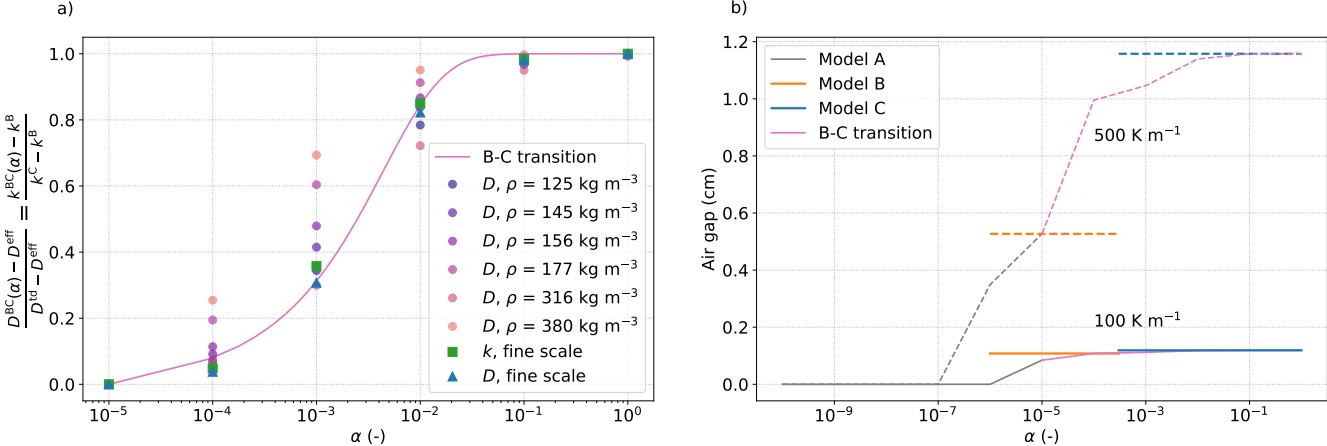

**Figure 17.** (a): Fitted B-C transition function Eq. (74) as a function of $\alpha$ for the simplified snow microstructure of Sect. 3.3 (solid pink line). Values from the pore-scale simulations, which were fitted to obtain the B-C transition, are shown by green squares for the thermal conductivity and by blue triangles for the diffusion coefficient. The circle markers show the diffusion coefficient estimates from Fourteau et al. (2021b) for different snow densities. (b): Estimation of the air gap simulated by the macroscopic models for the simplified microstructure after 15 days under a temperature gradient of 100 K m$^{-1}$ (solid lines) and of 500 K m$^{-1}$ (dotted lines). Results using the B-C transition are shown (pink lines).

In contrast to the model A, there is no smooth transition between the model B and the model C when varying the $\alpha$-value, as both do not depend on $\alpha$. Hence, at the first order, the temperature and vapor density fields obtained from those models apply for the entire $\alpha$ range of their domain of validity, which does not agree with the smooth evolution observed in the pore-scale simulation results when $\alpha$ varies in the range $[10^{-5}, 1]$ (see Fig. 11 and 12). As both models B and C present similar form, a way to reproduce this transition is to introduce a simple function to describe the evolution, when $\alpha$ varies, between the macroscopic parameters involved in the model B and in the model C, i.e. between $D^{\mathrm{eff}}$ and $D^{\mathrm{td}}$ and between $k^{\mathrm{B}}$ and $k^{\mathrm{C}}$. For that, we used the temperature and vapor density fields obtained from the pore-scale simulations (see Section 3.) over one REV of snow (the hundredth cell) at different $\alpha$-values to compute the apparent properties $D^{\mathrm{BC}}(\alpha)$ and $k^{\mathrm{BC}}(\alpha)$, which verify in first approximation the following relation:

$$\frac{D^{\mathrm{BC}}(\alpha) - D^{\mathrm{eff}}}{D^{\mathrm{td}} - D^{\mathrm{eff}}} \approx \frac{k^{\mathrm{BC}}(\alpha) - k^{\mathrm{B}}}{k^{\mathrm{C}} - k^{\mathrm{B}}} \approx \tanh\left(\left(\frac{\alpha - 10^{-5}}{b}\right)^{a}\right) \tag{74}$$

where $a = 0.58$ and $b = 0.00693$ are two constants. Figure 17.a shows the evolution of this function fitted on the numerical fine scale results of $D^{\mathrm{BC}}(\alpha)$ and $k^{\mathrm{BC}}(\alpha)$. The figure also includes the numerical estimations of the diffusion coefficient on 3D snow microstructure from Fourteau et al. (2021b), which are in good agreement with the proposed function. Using balance equations similar as the ones of the models B or C, but with the parameters $D^{\mathrm{BC}}(\alpha)$ and $k^{\mathrm{BC}}(\alpha)$, it is possible to compute the temperature and $\rho_{vs}(T)$ field in a refined way for $\alpha$-values in the range $[10^{-5}, 1]$. To illustrate, we applied this approach to the



simulations on the simplified 2D snow microstructure presented in Sect. 3.3 and the results are shown in Fig. 12. Temperature and vapor density from pore-scale simulations are closely reproduced by the macroscopic models in the $\alpha$ range $[10^{-5}, 1]$. The proposed fit (Eq. 74) also allows a continuous estimation of the air-gap as it is shown on Figure 17.b. The introduction of such a function can be useful for future studies working in the $\alpha$ range $[10^{-5}, 1]$, in which most experimental $\alpha$-values from the literature fall, as well as when running the model A in the range $[10^{-5}, \alpha_T]$, as it can present numerical instabilities above $10^{-5}$ (Schürholt et al., 2022).

### 5.2 On the comparisons between simulations and measurements

**Summary of the models' evaluation:** Comparing experiments and simulations with the three models, it appears that they are able to reproduce the main features of the heat and mass transport during TGM, including the non-linear temperature profile and, for the model A, the upward vapor transport with eventually the formation of a mm-scale basal air gap. However, a major discrepancy lies in the fact that temperature values are underestimated by all the models. More precisely, the heat source inducing the non-linearity in the temperature profile seems underestimated. The best predictions of the temperature deviation $\Delta T$ are obtained by the model C and correspond only about 25% of the experimental data, which translates into temperature differences of around 1 K and 5 K for the Bouvet A and Kamata experiments, respectively. To a much lesser extend, upward vapor transport seems slightly underestimated and the heights of the basal air gaps simulate by the model A corresponds about 75% of the experimental ones, leading to small differences in height of 1 mm and 0.6 mm for the experiment Bouvet B and Kamata, respectively. Similar conclusions seem to be drawn for the models B and C, based on rough approximations of the air gap. Possible causes of the differences between experiments and simulations are explored in the following.

**Uncertainties on the experimental data:** Temperature measurements in Bouvet A were performed with PT100 sensors with an accuracy of $\pm 0.2$°C (Bouvet et al., 2023). Copper-Constantan thermo-couples were used in the experiment of Kamata and Sato (2007) and are known to be very stable at low temperatures, with an accuracy of $\pm 0.5$°C. In both cases, these uncertainties are smaller than the discrepancies between the measured and modeled $\Delta T$. For density, the experimental setup of Bouvet B ensure precise monitoring of the mass change over time by tomography (Bouvet et al., 2023). Air gaps similar to the one in Bouvet B were reported by Wiese (2017) during temperature gradient experiments. For Kamata experiment, the reliability of the compartment method is less obvious, and the vertical resolution of 2.5 cm is rather poor to assess the presence of an air gap.

**Uncertainties on the numerical simulation input:** The macroscopic modeling of heat and mass transport in dry snow relies on the effective parameters $k^{\mathrm{eff}}$, $D^{\mathrm{eff}}$ and SSA, for the model A, on $k^{\mathrm{B}}$ for the model B, and on $k^{\mathrm{C}}$ for the model C. For thermal conductivity, the different estimates used in the simulations are overall in good agreement with the values computed on the experimental 3D images. A possible way of improvement could be to account for the anisotropy of the property, so for an enhanced thermal conductivity in the vertical direction as observed for snow evolving under a high TGM (e.g. Calonne et al., 2011). However, an increase in the thermal conductivity of the models A, B and C leads to a decrease in both temperature deviation and air gap height, and thus to degrade the models' performance, as illustrated in Fig. 15. Concerning the vapor diffusion





coefficient, the SC models was used in the simulations, which provide slightly overestimated estimates. However, improving these estimates, so taking lower values of the diffusion coefficient, leads to a decrease in both temperature deviation and air gap height, which, again, degrades the model results. To illustrate in the case of the model A, by lowering the SC estimate of $D^{\text{eff}}$ by 10%, a $\Delta T$ of 0.096 K and an air gap of 1.17 mm is simulated for $\alpha = 10^{-5}$, compared to 0.102 K and 2.3 mm with

the initial value. Finally, potential errors in the SSA parameter would affect the source term of the model A and would only translate in small variations of $\alpha$. To conclude, the uncertainties linked to the estimate of the effective parameters cannot be responsible for the reported differences between experiments and simulations.

**Models limitations and potential improvements:** As the points raised above do not seem sufficient to explain the models'
errors, a plausible cause remains to be investigated and is the definition of the models itself, i.e. the definition of the physics at the pore scale considered for the homogenization. A first element concerns the source terms in the model A, which are derived from the Hertz-Knudsen equation and relies on a condensation coefficient $\alpha$ (Eq. 7). Here, this coefficient was taken constant and uniform over the snow layer and considered equal for both condensation and sublimation. In their review, Persad and Ward (2016) explore the expressions of the evaporation coefficient and of the condensation coefficient in the Hertz-Knudsen
equation for the water-air interface. They conclude that most errors come from assuming the evaporation and condensation coefficients to be equal and assuming thermal equilibrium across the liquid-vapor interface (Eq. 4 in this study). Moreover, as mentioned in the introduction, the condensation parameter $\alpha$ depends on many parameters, such as the vapor supersaturation, which can lead to a non-linear expression of the Hertz-Knudsen equation. Hence, refining the Hertz-Knudsen equation could add non-linearity in the source terms of the model A, which could enhance the contribution of latent heat and thus increase the
temperature deviation $\Delta T$, which would improve the model's prediction.

Another point is that the natural convection was not taken into account at the pore scale. This process was however hypothesized to be key for heat and mass transport of snow under strong temperature gradients, such as Arctic and sub-Arctic ones (e.g. Sturm and Johnson, 1991; Domine et al., 2018). To include natural convection, fluxes of temperature ($J_T$) and water vapor ($J_{\rho_v}$) should be expressed at the pore-scale as follows:

$$J_T = -k_a\mathbf{grad}T_a + \rho_a C_a v_a T_a \quad \text{in } \Omega_a \quad \text{and} \quad J_{\rho_v} = -D_v\mathbf{grad}\rho_v + v_a\rho_v \quad \text{in } \Omega_a \tag{75}$$

with $v_a$ the air velocity. A numerical study was recently presented by Jafari et al. (2022) using a macroscopic model similar to the model A. They show that the occurrence and intensity of natural convection in snow depends on the Rayleigh number defined as:

$$Ra = \frac{\rho_a\beta_T g(T_{\text{bottom}} - T_{\text{top}})HK}{((\mu_a k^{\text{eff}})/(\rho_a C_a))} \tag{76}$$

where $H$ is the height of the snow layer, $K$ is the snow permeability, $g = 9.81 \text{ m s}^{-2}$ is the gravity, $\mu_a = 17.29 \times 10^{-6}$ Pa.s is the air viscosity and $\beta_T = 0.0036 \text{ K}^{-1}$ is the thermal expansion coefficient. Their simulations indicate that, for $Ra > 50$ and $H > 25$ cm, natural convection could generate an upward air flux from the warmer region to the colder one, and inversely. We estimated the Rayleigh number for the three experiments used in this study. Using the values in Table 1 and 2, and using the





parameterization of Calonne et al. (2012) for the snow permeability, the Rayleigh number is typically of 0.15, 0.02 and 0.85,
for the experiments Bouvet A, Bouvet B and Kamata, respectively. These values are much smaller than the threshold value
presented by Jafari et al. (2022), which would indicate that natural convection is negligible in our cases. Moreover, Kamata
et al. (1999) present a symmetric TGM experiment, with warmer conditions at the base and top of the snow layer and colder
conditions imposed in the middle using a cold plate. The snow layer was thus under a positive TG in one part and under a
negative TG in the other part, both of the same intensity. The temperature profiles were recorded in both parts of the snow layer
and show similar nonlinear curves in both cases, although natural convection could only occur in the bottom area, where the
temperature conditions are unstable. The authors conclude that natural convection does not seem to impact their temperature
fields. This would be consistent with the small Rayleigh number that we estimated to be 0.16 for this experiment.

Finally, the cross-coupling effects between the temperature and water vapor density, such as the Soret and Dufour effects,
were not considered in the physics at the pore scale. The effect of the vapor density gradient on the heat flux, called the
Dufour effect, is characterized by the diffusion-thermo coefficient $D^{\mathrm{Tv}}$, and the effect of temperature gradient on the vapor
density flux, called the Soret effect, is characterized by the thermo-diffusion coefficient $D^{\mathrm{vT}}$. Taking these effects into account,
temperature and water vapor flux can be expressed as follows:

$$J_T = -k_a\mathbf{grad}T_a - D^{\mathrm{Tv}}\mathbf{grad}\rho_v \quad \text{in } \Omega_a \quad \text{and} \quad J_{\rho_v} = -D_v\mathbf{grad}\rho_v - D^{\mathrm{vT}}\mathbf{grad}T_a \quad \text{in } \Omega_a \tag{77}$$

For porous media, the Dufour effect is neglected in most cases, whereas the Soret effect is often taken into account (e.g.,
Davarzani et al., 2010; Häussling Löwgren et al., 2020) and can be measured using the Soret coefficient defined as $S_T = D^{\mathrm{vT}}/D_v$. This coefficient is positive when the heaviest species in the pore space move toward the colder regions, and is
negative when they move toward the warmer regions. However, this coefficient could change sign when the temperature is
lowered (Chapman and Cowling, 1990; Caldwell, 1973). When the temperature is positive, the Soret coefficient for water
vapor is supposed to be positive. To the best of our knowledge, there is no data concerning this coefficient when temperature is
negative. The Soret effect can be easily introduced in pore-scale simulations in the case of the 2D simplified microstructure as
presented in Sect. 3.3. By doing so, we found that negative $S_T$ coefficients lead to increase the simulated temperature deviation
$\Delta T$, and inversely. For example, for $\alpha = 0.1$ and a temperature gradient of 500 K m$^{-1}$, a maximum value of $\Delta T$ of 6.5 K, 4.2
K and 2.8 K is simulated for $S_T$ equals to $-2 \times 10^{-4}$, 0 and $2 \times 10^{-4}$, respectively. The Soret effect can also be introduced in
the model C, by replacing $k_a + k_{\mathrm{diff}}$ by $k_a + k_{\mathrm{diff}} + S_T D_v L_{sg}/\rho_i$. Using the self consistent estimate of thermal conductivity
(Eq. 70) and for values of $S_T$ of $-2 \times 10^{-4}$, 0 and $2 \times 10^{-4}$, the maximum simulated values of $\Delta T$ for the Kamata experiment
are 6.3 K, 3.1 K and 1.95 K, respectively, whereas the experimental value is around 6 K. A negative Soret coefficient seems
thus suitable to improve the temperature simulations and better describe the experimental data. However, the influence of the
Soret effect on the air gap is not straightforward, as it seems to induce a downward movement of vapor molecules, thus opposed
to the formation of a basal air gap. These preliminary results show that the introduction of such coupling effects (Soret and/or
Dufour) between the temperature and the water vapor density in the modeling of heat and water vapor transfer in snowpacks is
interesting and that future works would be needed to investigate such an hypothesis.





# 6 Conclusion

This paper presents the definition and evaluation of the equivalent macroscopic modeling of heat and mass transport during TGM in dry snow. In a first part, we applied the homogenization process to retrieve the macroscopic models valid for condensation coefficients $\alpha$ ranging from $10^{-10}$ to 1. We showed that, at a transition value $\alpha_T \approx 3 \times 10^{-4}$, the modeling changes from vapor transport limited by sublimation-deposition (models A and B) to vapor transport limited by diffusion (model C). The homogenization process allowed to retrieve different models proposed in the literature (Calonne et al., 2014b; Hansen and Foslien, 2015; Moyne et al., 1988) and to clarify their domains of validity according to the $\alpha$-values. For $\alpha$ between $10^{-10}$ and $\alpha_T$, the model A consists of two equations of temperature and water vapor density coupled through the source terms, which are proportional to the Hertz Knudsen equation and therefore to $\alpha$. This model does not presume any assumption on the saturation of the vapor density. For $\alpha$ between $10^{-6}$ to $\alpha_T$, the model B can be seen as a particular case of the model A, when water vapor is saturated at the macroscopic scale. It consists of one temperature equation which does not involve $\alpha$ at the first order. Finally, for $\alpha$ between $\alpha_T$ and 1, the model C also consists of one temperature equation which does not involve $\alpha$, and the water vapor is at saturation. Although of the same form, this model differs strongly from the model B as it captures a new mechanism, namely the thermo-diffusion induced by the phase change. To ensure a continuous transition between the models B and C, a simple transition function depending on $\alpha$ is proposed.

In the second part of the paper, we evaluated the homogenized models A, B and C by comparing with three laboratory experiments of TGM of snow (Kamata and Sato, 2007; Bouvet et al., 2023), as well as by a numerical evaluation for a 2D simplified microstructure. Evaluations were performed based on the temperature and density profiles of snow, and more precisely, on the ability to reproduce two main features reported in the TGM experiments: the non-linear concave-shaped temperature profile, characterized by the temperature deviation from a linear gradient $\Delta T$, and the upward vapor transport leading to a mass loss or an air gap at the base of the snow layer. We showed that (i) the three models allow to reproduce the shape of the temperature profile but the values are largely underestimated, the best prediction being obtained with the model C and corresponding only to 25% of the experimental data; this major discrepancy highlights that a process that contributes to heat up the layer is not well captured, if at all, (ii) the model A allows to reproduce the upward vapor transport and the formation of a mm-scale basal air gap, the best result being obtained for the highest $\alpha$-value of $10^{-5}$, (iii) the models B and C do not allow to reproduce mass transport as they predict only mass gain in the snow layer, as they do not satisfy mass conservation in the present case. Potential improvements were suggested and include the refining or enrichment of the physic at the pore scale considered to derive the models, such as questioning the expression of the Hertz-Knudsen equation or the role of the Soret and/or Dufour effects, as well as improving the boundary conditions to allow for realistic mass transport for the models B and C.





*Data availability.* The theoretical development of the macroscopic equivalent descriptions are available in the Supplement.

*Author contributions.* CG, NC and LB conducted the derivation of the macroscopic models and LB the application of the models to the experimental data. The analyses and interpretations were carried out by LB, FF, NC and CG. LB and CG prepared the manuscript with 885 contributions from all co-authors.

*Competing interests.* The contact author has declared that none of the authors have any competing interests.

*Acknowledgements.* The 3SR lab is part of the Labex Tec 21 (Investissements d'Avenir, grant ANR-11-LABX0030). CNRM/CEN is part of Labex OSUG@2020 (Investissements d'Avenir, grant ANR-10-LABX-0056). This research has been supported by the Agence Nationale de la Recherche through the MiMESis-3D ANR project (ANR-19-CE01-0009).



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
