# Peer review of "Multiscale modeling of heat and mass transfer in dry snow: influence of the condensation coefficient and comparison with experiments"

_The Cryosphere, 2023_

## Referee Comment (RC1)

**Review of the manuscript titled "Multiscale modeling of heat and mass transfer in dry snow: influence of the condensation coefficient and comparison with experiments"**

The manuscript reports on the multiscale modeling of the heat and mass transfer in dry snow by using the homogenization technique to derive the macroscopic equations. The heat transfer is ruled by the conduction mechanism in ice and water vapor phases whereas the mass transfer of water vapor is described by the Fick's 2nd law. At the ice/fluid interface, the Hertz-Knudsen relation is used to describe the sublimation/deposition mechanism. For different order of magnitude of the sublimation/deposition rate, different cases (A, B1, B2 and C) are considered giving rise to different macroscopic models characterized by the effective coefficients. The details of the homogenization procedure are given in the supplementary document while the main results of the macroscopic models are summarized in the manuscript.

First, I verified the homogenization procedure in the "long" supplementary document for the cases A, B1, B2 and C. The notations are quite heavy with the superscript $\star$ for each term and it can be simplified. In general, I agree with these results except for the case B2 (see my comments below). For high order of magnitude of the sublimation/deposition rate (or high value of $\alpha$), this gives the same result obtained from the volume averaging method reported in Moyne et al. (1988) for the heat and mass transfer with condensation/evaporation problem in porous media.

The paper is interesting and well written. The development of the multiscale models is rigorous with a well-posed $\varepsilon$-models from the dimensional analysis. For this reason, I recommend the paper for publication after revision.

**Major comments:**

1. The authors should explicitly explain the choice of using the Hertz-Knudsen equation for describing the vapor flux at the solid/fluid interface.

   Instead of using an equilibrium condition at the solid/fluid interface as (the curvature effect is neglected)

$$
\begin{aligned}
k_i \boldsymbol{\nabla} T_i \cdot \mathbf{n} - k_a \boldsymbol{\nabla} T_a \cdot \mathbf{n} &= \frac{L_{sg}}{\rho_i} D_v \boldsymbol{\nabla} \rho_v \cdot \mathbf{n} \\
\rho_v &= \rho_{vs} \qquad \text{at } \Gamma_{fs},
\end{aligned} \tag{1}
$$

   the authors introduce Hertz-Knudsen law to take into account the non-equilibrium state for small value of $\alpha$. It means that an "complementary resistance" is added at the solid/fluid interface. This point should be clearly discussed at the beginning.

   Moreover, maybe it should be better to define the latent heat of sublimation by $L = L_{sg}/\rho_i$ in $J/kg$.

2. Dimensional analysis: the ratio of the heat conductivities of ice and air is

$$
[K] = \frac{k_{ic}}{k_{ac}} = 96 \simeq \mathcal{O}(\varepsilon^{-1}) \tag{2}
$$

   However, the authors assume that $[K] = \mathcal{O}(1)$. This point needs to be clarified.

3. I don't agree with the result of the model B2. From Eqs. B2.22 and B2.23 together with the periodicity condition, we have

$$\int_{\Gamma} (\rho_v^{(1)} - \rho_{vs}^{(1)}) dS = 0 \tag{3}$$

where $\rho_v^{(1)}(\mathbf{x}, \mathbf{y}, t)$ is periodic function depending on $\mathbf{x}$ and $\mathbf{y}$. This can not ensure that $\rho_v^{(1)} = \rho_{vs}^{(1)}$ on $\Gamma$ is a unique solution.

I suggest that the authors should find a solution for $\rho_v^{(1)} - \rho_{vs}^{(1)}$ by linearity as

$$\rho_v^{(1)} - \rho_{vs}^{(1)} = \boldsymbol{\chi} \cdot \boldsymbol{\nabla}_x T^{(0)} \tag{4}$$

combined with a solution for $\rho_{vs}^{(1)} = \gamma \mathbf{r}_a \cdot \boldsymbol{\nabla}_x T^{(0)}$, so that from Eqs. B2.22 and B2.23, we can obtain a consistent closure problem with a coupled term at the solid/fluid interface.

4. In my opinion, the effect of the sublimation/deposition needs to be better discussed in the macroscopic results in the Section 2.6. For example, for the case C, what I understand is that considering the sublimation/deposition at the solid/fluid interface refers to a classical heat conduction problem for ice and air without sublimation/deposition with a modified air conductivity being $k_a + k_{dif}$.

5. I find that the discontinuity between the models B and C is quite surprising. Let consider only a heat conduction problem with a resistance at the solid/fluid interface as reported in Auriault et al. [1]. All the one equation models can be deduced from one to other. The discontinuity appears when passing from two equations models to one equation model. However in this work, the one equations models are not continuous. By revisiting the model B2 (see my comment 3), can we obtain the continuity of the models?

6. Page 17, line 410: it was concluded that if a temperature gradient is applied along $\mathbf{e}_2$, the model A (or B) will not predict any mass variation.

In this direction, $D_{22}^{eff} = 0$ and at the steady state, we have $\rho_v^{(0)} = \rho_{vs}^{(0)}(T^{(0)})$ which varies according to the Clausius Clapeyron's law.

7. Page 21, line 470: for the water vapor boundary conditions at the top and bottom, why the Robin boundary condition is imposed instead of using the zero-flux as applied for the macro-scale simulations?

8. Comparison between DNS and macroscopic simulations: In Figs. 11(a) and (b) for $\Delta T$, we observe clearly that by increasing $\alpha$, the DNS result tends to the one of model C and for higher value of $\alpha$ ($\alpha > 1$), we may have a good agreement between the DNS and the model C as expected. However, in Fig. 11(f), why the result of the DNS for $\alpha \to 1$ does not tend to the case C for $\dot{\phi}$?

Moreover, as the model C is independent on $\alpha$, I suggest that to compare with the simulation of the model C, for the mass transfer problem at the pore scale, the equilibrium should be used at the solid/fluid interface $\rho_v = \rho_{vs}$ at $\Gamma_{fs}$, instead of using the Robin condition involving the parameter $\alpha$.

9. It is observed that the model B and C can not predict correctly the behavior of sublimation/deposition in the vicinity of the boundary, in comparing with the DNS. In my opinion, it refers to a boundary layer problem (several works in the literature try to fix this problem encountered in simulation of homogenized models).

10. The terminology "boundary condition" used to describe the solid/fluid interface condition is not correct. Please modify this sentence to "interface condition".

**References**

[1] J. -L. Auriault, H. I. Ene, Macroscopic modelling of heat transfer in composites with interfacial thermal barrier, *Int. J. Heat Mass Transfer*, **37** (1994).

---

## Author Comment (AC1)

**Authors' reply to referee comments RC2 of the paper tc-2023-148 entitled "Multiscale modeling of heat and mass transfer in dry snow: influence of the condensation coefficient and comparison with experiments" by Bouvet et al.**

We thank very much Reviewer 2 for the comments. Please find below our point-by-point replies in blue color.

In the wake of a previous study of Calonne et al. (2014b), this paper aims presents a multiscale approach to follow heat and mass transfers in dry snow focusing on the peculiar role of the condensation coefficient $\alpha$ in order to mimic natural snow evolution during changes in the snow microstructure called temperature gradient metamorphism (TGM). Using a two-fold homogenization of a model coupling the heat conduction through ice and air, the water vapor diffusion in air and the sublimation of ice and deposition of vapor at the ice grain interface, this study's interest mainly consists in the fact that the effect of the condensation coefficient $\alpha$ is poorly described in the literature. Thus, considering a large range for this parameter (from 10-10 to 1), different effective behaviors are obtained through the upscaling process according to a transition value $\alpha$T $\approx$ 3×10-4. Moreover, the homogenized modelling results were compared with three experimental tests of TGM of snow, providing a solid discussion.

In general, the manuscript is well-written and constructed, and the mathematical statement, the upscaling procedure and the experimental comparisons are well presented. The quality of this paper is indubitable, and I only have minor concerns.

i/ The main articulation between this study and the previous results of the group on the same topic should be highlighted. Indeed, even if many new results are presented here, some interesting links can be made.

The present work follows previous work of Calonne et al. (2014) and Calonne et al. (2015) from our group, as well as other papers from the literature, but explores the full range of condensation coefficient values, broadening the understanding of heat and mass transport modeling in dry snow. This is outlined in the introduction of the paper. The Model A presented in the manuscript corresponds to the model proposed in Calonne et al. (2014), while the Model D (in the revised version) corresponds to the models of Moyne et al. (1988) and Hansen and Foslien (2015). The description of each model and their comparison with existing models are provided in Section 4.2 and recalled in the paper's conclusion.

ii/ The consequences of the model's assumptions (isotropic materials properties, no convective effect, no curvature effects, no natural convection at the pore scale, etc.) could be outlined. In particular, the use of the Hertz-Knudsen at the fluid-solid interphase is one of the key point of this model and requires to be justified in this context.

At the microscopic scale, we have assumed in first approximation that all the material properties are isotropic. This is true for all the properties related to the air, and it seems also reasonable for the properties related to the ice skeleton, such as its thermal properties, as the ice skeleton is constituted of an assemblage of single crystals with isotropic thermal properties. The condensation coefficient $\alpha$ arising in the Hertz-Knudsen is taken as isotropic, although it has been shown that it depends on the ice crystalline orientation. However, since the values of $\alpha$ remain of the same order of magnitude (see experimental dataset from e.g. Libbrecht and Rickerby (2013)), this hypothesis has no consequence on the present analysis.

Concerning the convection, in the present work for the sake of simplicity, we assumed that there is no forced convection induced by wind pumping for example. In the future, such convection could be included in the modeling as it has been done in Calonne et al. (2015) in order to have a more comprehensive model. The natural convection is also neglected in the present analysis. Even if such convection can be present in some particular situations, as it has been simulated recently by Jafari et al. (2022), the model predictions are here compared to experimental data for which natural convection is negligible, as discussed in the manuscript in section 5.2. Consequently, our concluding remarks are not affected by such hypothesis.

Finally, we have considered that the condensation-sublimation processes arising at the ice-air interface are driven by the Hertz-Knudsen equation. This latter equation, initially derived to describe the condensation-evaporation processes at a liquid-gas interface, is widely used in snow physics and is supported by several experimental evidences (e.g., Libbrecht, 2005; Kaempfer and Plapp, 2009; Libbrecht and Rickerby, 2013; Furukawa, 2015; Krol and Löwe, 2016). Moreover, to the best

of our knowledge, we are not aware of any another model to describe such processes. The use of classical models encountered in solidification does not seem suitable. The above considerations about the choice of the Hertz-Knudsen equation were more clearly stated in the revised version of the manuscript (line 118).

iii/ The evaluation of the dimensionless numbers defined by Eq. (19) is a key point of the upscaling procedure. Thus if the sensitivity to the coefficient $\alpha$ is well introduced, one may wonder if other similar dependencies of some dimensionless numbers (according to the temperature for instance) may not be discussed.

Most of the physical parameters involved in the problem at the pore scale are temperature dependent, but only to a small extent. Variations in the physical parameters due to temperature dependence cannot change the order of magnitude of the dimensionless numbers in terms of $\varepsilon$, and thus the derived macroscopic models.

Notwithstanding these general remarks, this is a complete work coupling models, numerical simulation and experimental comparisons. It is rigorously presented and detailed in the appendix. That is why, if these minor suggestions are addressed, I suggest to accept the publication of this work.

**References**

Calonne, N., Geindreau, C., and Flin, F.: Macroscopic modeling for heat and water vapor transfer in dry snow by homogenization, The Journal of Physical Chemistry B, 118, 13 393–13 403, https://doi.org/10.1021/jp5052535, 2014.

Calonne, N., Geindreau, C., and Flin, F.: Macroscopic modeling of heat and water vapor transfer with phase change in dry snow based on an upscaling method: Influence of air convection, Journal of Geophysical Research: Earth Surface, 120, 2476–2497, https://doi.org/10.1002/2015JF003605, 2015.

Furukawa, Y.: 25 - Snow and Ice Crystal Growth, in: Handbook of Crystal Growth (Second Edition), edited by Nishinaga, T., pp. 1061–1112, Elsevier, Boston, second edition edn., https://doi.org/https://doi.org/10.1016/B978-0-444-56369-9.00025-3, 2015.

Hansen, A. C. and Foslien, W. E.: A macroscale mixture theory analysis of deposition and sublimation rates during heat and mass transfer in dry snow, The Cryosphere, 9, 1857–1878, https://doi.org/10.5194/tc-9-1857-2015, 2015.

Jafari, M., Sharma, V., and Lehning, M.: Convection of water vapour in snowpacks, Journal of Fluid Mechanics, 934, A38, https://doi.org/10.1017/jfm.2021.1146, 2022.

Kaempfer, T. U. and Plapp, M.: Phase-field modeling of dry snow metamorphism, Phys. Rev. E, 79, 031 502, https://doi.org/10.1103/PhysRevE.79.031502, 2009.

Krol, Q. and Löwe, H.: Relating optical and microwave grain metrics of snow: The relevance of grain shape, The Cryosphere, 10, 2847–2863, https://doi.org/10.5194/tc-10-2847-2016, 2016.

Libbrecht, K. G.: The physics of snow crystals, Rep. Prog. Phys., 68, 855–895, https://doi.org/doi:10.1088/0034-4885/68/4/R03, 2005.

Libbrecht, K. G. and Rickerby, M. E.: Measurements of surface attachment kinetics for faceted ice crystal growth, Journal of Crystal Growth, 377, 1–8, https://doi.org/10.1016/j.jcrysgro.2013.04.037, 2013.

Moyne, C., Batsale, J.-C., and Degiovanni, A.: Approche expérimentale et théorique de la conductivité thermique des milieux poreux humides—II. Théorie, International Journal of Heat and Mass Transfer, 31, 2319–2330, https://doi.org/10.1016/0017-9310(88)90163-9, 1988.

---

## Author Comment (AC2)

**Authors' reply to referee comments RC1 of the paper tc-2023-148 entitled "Multiscale modeling of heat and mass transfer in dry snow: influence of the condensation coefficient and comparison with experiments" by Bouvet et al.**

We thank very much Reviewer 1 for his fruitful comments. Please find below our point-by-point replies in blue color.

The manuscript reports on the multi-scale modeling of the heat and mass transfer in dry snow by using the homogenization technique to derive the macroscopic equations. The heat transfer is ruled by the conduction mechanism in ice and water vapor phases whereas the mass transfer of water vapor is described by the Fick's 2nd law. At the ice/fluid interface, the Hertz-Knudsen relation is used to describe the sublimation/deposition mechanism. For different order of magnitude of the sublimation/deposition rate, different cases (A, B1, B2 and C) are considered giving rise to different macroscopic models characterized by the effective coefficients. The details of the homogenization procedure are given in the supplementary document while the main results of the macroscopic models are summarized in the manuscript. First, I verified the homogenization procedure in the "long" supplementary document for the cases A, B1, B2 and C. The notations are quite heavy with the superscript ? for each term and it can be simplified. In general, I agree with these results except for the case B2 (see my comments below). For high order of magnitude of the sublimation/deposition rate (or high value of $\alpha$), this gives the same result obtained from the volume averaging method reported in Moyne et al. (1988) for the heat and mass transfer with condensation/evaporation problem in porous media. The paper is interesting and well written. The development of the multiscale models is rigorous with a well-posed $\varepsilon$-models from the dimensional analysis. For this reason, I recommend the paper for publication after revision. Major comments:

1. The authors should explicitly explain the choice of using the Hertz-Knudsen equation for describing the vapor flux at the solid/fluid interface. Instead of using an equilibrium condition at the solid/fluid interface as (the curvature effect is neglected)

$$k_i \boldsymbol{\nabla} T_i \cdot \mathbf{n} - k_a \boldsymbol{\nabla} T_a \cdot \mathbf{n} = \frac{L_{sg}}{\rho_i} D_v \boldsymbol{\nabla} \rho_v \cdot \mathbf{n} \tag{1}$$

$$\rho_v = \rho_{vs} \quad \text{at } \Gamma_{fs}, \tag{2}$$

the authors introduce Hertz-Knudsen law to take into account the non-equilibrium state for small value of $\alpha$. It means that an "complementary resistance" is added at the solid/fluid interface. This point should be clearly discussed at the beginning. Moreover, maybe it should be better to define the latent heat of sublimation by $L = L_{sg}/\rho_i$ in J/kg.

In the present work, we have considered that the condensation-sublimation processes arising at the ice-air interface are driven by the Hertz-Knudsen equation. This latter equation, initially derived to describe the condensation-evaporation processes at a liquid-gas interface, is widely used in snow physics and is supported by several experimental evidences (e.g., Libbrecht, 2005; Kaempfer and Plapp, 2009; Furukawa, 2015; Libbrecht and Rickerby, 2013; Krol and Löwe, 2016). The Hertz-Knudsen equation is linearly dependent on the condensation coefficient $\alpha$, which characterizes the probability (theoretically ranges from 0 to 1 for an infinite flat surface) that a water vapor molecule striking the ice surface is incorporated into it (see e.g., Libbrecht, 2005; Furukawa, 2015). As suggested by the reviewer, this coefficient $\alpha$ can be also seen as a "complementary resistance" at the solid /fluid interface. The above considerations were included in the revised version of the manuscript (line 118) to better explain the choice of the Hertz-Knudsen equation.

We agree with the suggestion of the reviewer that defining the latent heat of sublimation by $L_{sg}^\rho = L_{sg}/\rho_i$ in J/kg would simplify the expression thorough the paper. However, for the sake of consistency with our previous works (Bouvet et al., 2023; Calonne et al., 2014, 2015), we prefer to keep the same notation.

2. Dimensional analysis: the ratio of the heat conductivities of ice and air is

$$[K] = \frac{k_{ic}}{k_{ac}} = 96 \simeq \mathcal{O}\left(\varepsilon^{-1}\right) \tag{3}$$

However, the authors assume that [K] $=\mathcal{O}(1)$. This point needs to be clarified.

We agree with the reviewer that [K]$= \mathcal{O}\left(\varepsilon^{-1}\right)$ if we assume that $\varepsilon = 5 \times 10^{-3}$, i.e. $l \approx 5 \times 10^{-4}$ m and $L \approx 0.1$ m. If we assume that [K]$= \mathcal{O}\left(\varepsilon^{-1}\right)$, it is equivalent to neglect the effect of the thermal conductivity of the air on the effective thermal conductivity. In the analysis, we have kept [K] $=\mathcal{O}(1)$ for two reasons: (i) to keep the influence of the air on heat transfer for the case of extremely porous snow, and (ii) to ensure the continuity of the thermal conductivity at the boundary between the ice and the air phase. Moreover, let us remark that, in practice, $\varepsilon$ may be of the order $10^{-4}$ (for small grains, $l \sim 0.1$ mm and a snowpack thickness $L = 1$ m), in that case [K] $=\mathcal{O}(1)$.

3. I don't agree with the result of the model B2. From Eqs. B2.22 and B2.23 together with the periodicity condition, we have

$$\int_{\Gamma} \left(\rho_v^{(1)} - \rho_{vs}^{(1)}\right) dS = 0 \tag{4}$$

where $\rho_v^{(1)}(x, y, t)$ is periodic function depending on x and y. This can not ensure that $\rho_v^{(1)} = \rho_{vs}^{(1)}$ on $\Gamma$ is a unique solution. I suggest that the authors should find a solution for $\rho_v^{(1)} - \rho_{vs}^{(1)}$ by linearity as

$$\rho_v^{(1)} - \rho_{vs}^{(1)} = \chi \cdot \nabla_x T^{(0)} \tag{5}$$

combined with a solution for $\rho_{vs}^{(1)} = \gamma \mathbf{r}_a \cdot \nabla_x T^{(0)}$, so that from Eqs. B2.22 and B2.23, we can obtain a consistent closure problem with a coupled term at the solid/fluid interface.

We agree with this comment, and substantial modifications were made accordingly in the revised version of the manuscript. Taking into account the remarks of the reviewer, we have revised the derivation of the models (B2, C1). These modifications give rise to a new model called "C" in the revised version of the article. The model "C" ensures the continuity between the model B and D (the model D of the new version of the manuscript corresponds to the model C in the initial version) when $[\mathrm{W_R}] = \mathcal{O}(1)$, i.e $\varepsilon^{1/2} < [\mathrm{W_R}] < \varepsilon^{-1/2}$ and thus when $\alpha_{\min} = (\varepsilon^{1/2} D_{vc}/(l w_{kc})) < \alpha < \alpha_{\max} = (\varepsilon^{-1/2} D_{vc}/(l w_{kc}))$. In dimensionless form, this model is written:

$$(\rho C)^{\mathrm{eff}*} \frac{\partial T^{*(0)}}{\partial t^*} - \mathrm{div}_{x^*}(\mathbf{k}^{\mathrm{C}*}\mathbf{grad}_{x^*} T^{*(0)}) = L_{sg}^* \dot{\phi} \tag{6}$$

$$\phi \frac{\partial \rho_{vs}^{*(0)}}{\partial t} - \mathrm{div}_{x^*}(\mathbf{D}^{\mathrm{C}*}\mathbf{grad}_{x^*} \rho_{vs}^{*(0)}(T^{*(0)})) = \rho_i^* \dot{\phi} \tag{7}$$

where $(\rho C)^{\mathrm{eff}*}$ and $\mathbf{k}^{\mathrm{C}*}$ are the dimensionless effective thermal capacity and the effective dimensionless thermal conductivity respectively, defined as:

$$(\rho C)^{\mathrm{eff}*} = (1 - \phi)\rho_i^* C_i^* + \phi \rho_a^* C_a^* \tag{8}$$

$$\mathbf{k}^{\mathrm{C}*} = \frac{1}{|\Omega|} \left( \int_{\Omega_a} k_a^*(\mathbf{grad}_{y^*} \mathbf{s}_a^* + \mathbf{I}) d\Omega + \int_{\Omega_i} k_i^*(\mathbf{grad}_{y^*} \mathbf{s}_i^* + \mathbf{I}) d\Omega \right) \tag{9}$$

and where $\mathbf{D}^{\mathrm{C}*}$ is the dimensionless effective diffusion tensor defined as:

$$\mathbf{D}^{\mathrm{C}*} = \frac{1}{|\Omega|} \int_{\Omega_a} D_v^*(\mathbf{grad}_{y^*}(\mathbf{d}^* + \mathbf{s}_a^*) + \mathbf{I}) d\Omega \tag{10}$$

where $\mathbf{s}_i^*(\mathbf{y}^*)$, $\mathbf{s}_a^*(\mathbf{y}^*)$ and $\mathbf{d}^*(\mathbf{y}^*)$ are periodic vectors solution of the following coupled boundary value problem in a compact form:

$$\text{div}_{y^*}(k_i^*(\mathbf{grad}_{y^*}\mathbf{s}_i^* + \mathbf{I})) = 0 \quad \text{in } \Omega_i \tag{11}$$

$$\text{div}_{y^*}(k_a^*(\mathbf{grad}_{y^*}\mathbf{s}_a^* + \mathbf{I})) = 0 \quad \text{in } \Omega_a \tag{12}$$

$$\mathbf{s}_i^* = \mathbf{s}_a^* \quad \text{on } \Gamma \tag{13}$$

$$(k_i^*(\mathbf{grad}_{y^*}\mathbf{s}_i^* + \mathbf{I}) - k_a^*(\mathbf{grad}_{y^*}\mathbf{s}_a^* + \mathbf{I})) \cdot \mathbf{n_i} = \frac{L_{sg}^*}{\rho_i^*}\alpha^* w_k^* \gamma^*(T^{*(0)})\mathbf{d}^* \quad \text{on } \Gamma \tag{14}$$

$$\text{div}_{y^*}(D_v^*(\mathbf{grad}_{y^*}(\mathbf{d}^* + \mathbf{s}_a^*) + \mathbf{I})) = 0 \quad \text{in } \Omega_a \tag{15}$$

$$D_v^*(\mathbf{grad}_{y^*}(\mathbf{d}^* + \mathbf{s}_a^*) + \mathbf{I}) \cdot \mathbf{n_i} = \alpha^* w_k^* \mathbf{d}^* \quad \text{on } \Gamma \tag{16}$$

with

$$\frac{1}{|\Omega|}\int_\Omega (\mathbf{s}_a^* + \mathbf{s}_i^*)\mathrm{d}\Omega = \mathbf{0} \tag{17}$$

$$\frac{1}{|\Omega|}\int_\Gamma \mathbf{d}^*\mathrm{d}\Gamma = \mathbf{0} \tag{18}$$

The vector $\mathbf{s}_i^*$, $\mathbf{s}_a^*$ and $\mathbf{d}^*$ depend on the value of $\alpha$ and of the temperature (notably through $\gamma^*(T^{*(0)})$). The evolution of the parameters $\mathbf{k}^C$ and $\mathbf{D}^C$ with respect to $\alpha$ is illustrated below in Figure 1 for the particular case of the simplified 2D geometry presented in the section 3.3 of the manuscript, assuming that $k_i$ and $k_a$ are constant. This new figure was also added in the new version of the manuscript (Figure 10). The following description of the figure was also included line 550: 'Figure 10 shows the evolution of the dimensionless diffusion coefficients $D^{\text{eff}}/D_v$, $D^C/D_v$ and $D^D/D_v$ and of the macroscopic thermal conductivities $k^{\text{eff}}$, $k^C$ and $k^D$ with respect to $\alpha$ and for two temperatures of 270K and 250K. As expected, only the parameters of the model C ($D^C/D_v$ and $k^C$) vary with $\alpha$ and ensure a continuous transition between the parameters of the model A ($D^{\text{eff}}/D_v$ and $k^{\text{eff}}$) and the ones of the model D ($D^D/D_v$ and $k^D$). Fitting the numerical estimates, such transition can be described by a simple function:

$$\frac{D^C(\alpha) - D^{\text{eff}}}{D^D - D^{\text{eff}}} = \frac{k^C(\alpha) - k^{\text{eff}}}{k^D - k^{\text{eff}}} = \frac{\tilde{k}^C(\alpha) - \tilde{k}^B}{\tilde{k}^D - \tilde{k}^B} = \frac{A\alpha}{1 + A\alpha}$$

where $A = 1200$ is a constant.'

4. In my opinion, the effect of the sublimation/deposition needs to be better discussed in the macroscopic results in the Section 2.6. For example, for the case C, what I understand is that considering the sublimation/deposition at the solid/fluid

[Figure]

**Figure 1.** Evolution of the dimensionless diffusion coefficients $D^{\mathrm{eff}}/D_v$, $D^{\mathrm{C}}/D_v$ and $D^{\mathrm{D}}/D_v$ and of the macroscopic thermal conductivities $k^{\mathrm{eff}}$, $k^{\mathrm{C}}$ and $k^{\mathrm{D}}$ with respect to $\alpha$ and for two temperatures (270K and 250K). The black lines represents the proposed function Eq. (89) to describe the parameters of the model C. Numerical estimates of the diffusion coefficient on 3D snow microstructures of different densities from Fourteau et al. (2021) are represented by the dot symbols.

interface refers to a classical heat conduction problem for ice and air without sublimation/deposition with a modified air conductivity being $k_a + k_{\mathrm{dif}}$.

When $\alpha$ tends towards 1, i.e. $[\mathrm{W_R}] > \mathcal{O}(1)$, we have shown that $[\mathrm{H}] = k_{\mathrm{dif}_c}/k_{a_c}$ (see section 2.4) and consequently the microscopic problem becomes a classical heat conduction problem. Thus, in that case, the sublimation/deposition process induces at the first order an increase of the air conductivity through $k_{\mathrm{dif}}$, which is proportional to the latent heat. This result is consistent with the results obtained from the volume averaging method reported in Moyne et al. (1988).

5. I find that the discontinuity between the models B and C is quite surprising. Let consider only a heat conduction problem with a resistance at the solid/fluid interface as reported in Auriault and Ene (1994). All the one equation models can be deduced from one to other. The discontinuity appears when passing from two equations models to one equation model. However in this work, the one equations models are not continuous. By revisiting the model B2 (see my comment 3), can we obtain the continuity of the models?

Thanks to the remark of the reviewer (comment 3), a new model "C" has been derived, which ensures now the continuity between the models B and D, as presented in our reply of comment 3. This new model "C" was included in the revised version of the manuscript.

6. Page 17, line 410: it was concluded that if a temperature gradient is applied along e2, the model A (or B) will not predict any mass variation. In this direction, $D_{22}^{eff} = 0$ and at the steady state, we have $\rho_v^{(0)} = \rho_{vs}^{(0)}\left(T^{(0)}\right)$ which varies according to the Clausius Clapeyron's law.

We wanted to point out that in that case there is no variation of porosity, i.e of the snow density. Indeed in the case of the model B, according to the equation (37), in steady state condition, we have $\dot{\phi} = 0$. (since $\mathbf{grad}\rho_{vs}^{(0)} = \gamma(T^{(0)})\mathbf{grad}T^{(0)}$ is along $\mathbf{e}_2$ only). This part has been changed accordingly in the revised version of the manuscript.

7. Page 21, line 470: for the water vapor boundary conditions at the top and bottom, why the Robin boundary condition is imposed instead of using the zero-flux as applied for the macro-scale simulations?

We agree with this remark. Simulations at the micro-scale were thus performed again and a zero-flux was applied. The
130   obtained results do not differ significantly from the previous applied conditions, as shown by the figures below (Fig. 2 and 3). These updated figures were included in the revised version of the manuscript (Figure 12 and 13).

8. Comparison between DNS and macroscopic simulations: In Figs. 11(a) and (b) for $\Delta T$ , we observe clearly that by increasing $\alpha$, the DNS result tends to the one of model C and for higher value of $\alpha$ ($\alpha > 1$), we may have a good agreement between the DNS and the model C as expected. However, in Fig. 11(f), why the result of the DNS for $\alpha \longrightarrow$
135   1 does not tend to the case C for $\dot{\Phi}$? Moreover, as the model C is independent on $\alpha$, I suggest that to compare with the simulation of the model C, for the mass transfer problem at the pore scale, the equilibrium should be used at the solid/fluid interface $\rho_v = \rho_{vs}$ at $\Gamma$, instead of using the Robin condition involving the parameter $\alpha$.

The figures 2 and 3 presented here were updated in the revised version of the manuscript and correspond to Figures 12 and 13 (figures 11 and 12 in the initial manuscript). They take into account the new results from the micro-scale
140   simulations (see previous comment) as well as the predictions of the new model C. These figures show that a better agreement between the model predictions and the fine scale simulations occurs when $\alpha$ tends towards 1. However, we can still observe some differences for $\alpha$-values in the range $10^{-5} - 1$, which may be due to the inaccuracy of the mass balance equation. A boundary layer must be introduced to better predict sublimation/deposition in the vicinity of the bottom and top boundaries.

145

9. It is observed that the model B and C can not predict correctly the behavior of sublimation/deposition in the vicinity of the boundary, in comparing with the DNS. In my opinion, it refers to a boundary layer problem (several works in the literature try to fix this problem encountered in simulation of homogenized models).

We agree with the reviewer that a boundary layer can be introduced, as in the case of heterogeneous reactions in porous
150   media, at the bottom and the top of the snowpack to better predict sublimation/deposition in the vicinity of these boundaries. This problem will be addressed in future works.

10. The terminology "boundary condition" used to describe the solid/fluid interface condition is not correct. Please modify this sentence to "interface condition".

155   In the revised version of the manuscript, the terminology "boundary condition" has been replaced by "interface condition" as suggested by the reviewer.

[Figure]

**Figure 2.** Vertical profiles of $\Delta T$, $\rho - \rho_{vs}(T)$, and $\dot{\phi}$ from the pore scale simulations (dots) and from the macroscopic model A (grey lines), B (orange lines), C (magenta lines) and D (blue lines), considering a temperature gradient of 100 and 500 K m$^{-1}$ and for different values of $\alpha$. $\Delta T$ represents the deviation of the temperature profile from a linear temperature profile.

[Figure]

**Figure 3.** Temperature and water vapor density in the middle of the snow layer as a function of $\alpha$, obtained from the pore-scale simulations (dots) and from the macroscopic model A (grey lines), B (orange lines), C (magenta lines) and D (blue lines), at 100 and 500 K m$^{-1}$. The models are only shown for the $\alpha$-values within their domain of validity. Values of saturation water vapor density $\rho_{vs}$ from the pore-scale simulations and from the model A are also presented.

**References**

Auriault, J.-L. and Ene, H. I.: Macroscopic modelling of heat transfer in composites with interfacial thermal barrier, International journal of heat and mass transfer, 37, 2885–2892, 1994.

Bouvet, L., Calonne, N., Flin, F., and Geindreau, C.: Heterogeneous grain growth and vertical mass transfer within a snow layer under temperature gradient, The Cryosphere Discussions, 2023, 1–30, https://doi.org/10.5194/tc-2022-255, 2023.

Calonne, N., Geindreau, C., and Flin, F.: Macroscopic modeling for heat and water vapor transfer in dry snow by homogenization, The Journal of Physical Chemistry B, 118, 13 393–13 403, https://doi.org/10.1021/jp5052535, 2014.

Calonne, N., Geindreau, C., and Flin, F.: Macroscopic modeling of heat and water vapor transfer with phase change in dry snow based on an upscaling method: Influence of air convection, Journal of Geophysical Research: Earth Surface, 120, 2476–2497, https://doi.org/10.1002/2015JF003605, 2015.

Fourteau, K., Domine, F., and Hagenmuller, P.: Macroscopic water vapor diffusion is not enhanced in snow, The Cryosphere, 15, 389–406, https://doi.org/10.5194/tc-15-389-2021, 2021.

Furukawa, Y.: 25 - Snow and Ice Crystal Growth, in: Handbook of Crystal Growth (Second Edition), edited by Nishinaga, T., pp. 1061–1112, Elsevier, Boston, second edition edn., https://doi.org/https://doi.org/10.1016/B978-0-444-56369-9.00025-3, 2015.

Kaempfer, T. U. and Plapp, M.: Phase-field modeling of dry snow metamorphism, Phys. Rev. E, 79, 031 502, https://doi.org/10.1103/PhysRevE.79.031502, 2009.

Krol, Q. and Löwe, H.: Relating optical and microwave grain metrics of snow: The relevance of grain shape, The Cryosphere, 10, 2847–2863, https://doi.org/10.5194/tc-10-2847-2016, 2016.

Libbrecht, K. G.: The physics of snow crystals, Rep. Prog. Phys., 68, 855–895, https://doi.org/doi:10.1088/0034-4885/68/4/R03, 2005.

Libbrecht, K. G. and Rickerby, M. E.: Measurements of surface attachment kinetics for faceted ice crystal growth, Journal of Crystal Growth, 377, 1–8, https://doi.org/10.1016/j.jcrysgro.2013.04.037, 2013.

Moyne, C., Batsale, J.-C., and Degiovanni, A.: Approche expérimentale et théorique de la conductivité thermique des milieux poreux humides—II. Théorie, International Journal of Heat and Mass Transfer, 31, 2319–2330, https://doi.org/10.1016/0017-9310(88)90163-9, 1988.

---

## Editor Decision (ED1)

[revised manuscript text omitted]
}_{\mathrm{SC}}^{\mathrm{eff}}, \mathbf{D}_{\mathrm{SC}}^{\mathrm{eff}}, \mathrm{SSA}_{\mathrm{Fit}}(t)$ |
| | Set Calonne | $\mathbf{k}_{\mathrm{Calonne}}^{\mathrm{eff}}, \mathbf{D}_{\mathrm{SC}}^{\mathrm{eff}}, \mathrm{SSA}_{\mathrm{Fit}}(t)$ |
| Model B | Set SC | $\tilde{\mathbf{k}}_{\mathrm{SC}}^{\mathrm{B}} = \mathbf{k}_{\mathrm{SC}}^{\mathrm{eff}} + k_{\mathrm{dif}}\mathbf{D}_{\mathrm{SC}}^{\mathrm{eff}}/D_v, \mathbf{D}_{\mathrm{SC}}^{\mathrm{eff}}$ |
| | Set Calonne | $\tilde{\mathbf{k}}_{\mathrm{Calonne}}^{\mathrm{B}} = \mathbf{k}_{\mathrm{Calonne}}^{\mathrm{eff}} + k_{\mathrm{dif}}\mathbf{D}_{\mathrm{SC}}^{\mathrm{eff}}/D_v, \mathbf{D}_{\mathrm{SC}}^{\mathrm{eff}}$ |
| Model C | Set SC | $\tilde{\mathbf{k}}_{\mathrm{SC}}^{\mathrm{C}}(\alpha)$ and $\mathbf{D}_{\mathrm{SC}}^{\mathrm{C}}(\alpha)$ from Eq. (89) |
| | Set Calonne | $\tilde{\mathbf{k}}_{\mathrm{Calonne}}^{\mathrm{C}}(\alpha)$ and $\mathbf{D}_{\mathrm{Calonne}}^{\mathrm{C}}(\alpha)$ from Eq. (89) |
| Model D | Set SC | $\tilde{\mathbf{k}}_{\mathrm{SC}}^{\mathrm{D}}, \mathbf{D}_{\mathrm{SC}}^{\mathrm{D}}$ |
| | Set Fit | $\tilde{\mathbf{k}}_{\mathrm{Fit}}^{\mathrm{D}}, \mathbf{D}_{\mathrm{SC}}^{\mathrm{
[revised manuscript text omitted]